# Mineralogical Properties of the Copper Slags from the SarCheshmeh Smelter Plant, Iran, in View of Value Recovery

**Saeed Mohamadi Nasab [1], Behnam Shafiei Bafti [1,*], Mohamad Reza Yarahmadi [2], Mohammad Mahmoudi Maymand [2] and Javad Kamalabadi Khorasani [2]**

[1] Department of Geology, Faculty of Science, Shahid Bahonar University of Kerman, Kerman 7616913439, Iran
[2] Research and Development Division, SarCheshmeh Copper Complex, Rafsanjan 1513744513, Iran
* Correspondence: behnam.shafiei@gmail.com or shafieibaftibehnam@uk.ac.ir; Tel.: +98-91-1269-1236

**Abstract:** Annually, hundreds of thousands of tons of slags are involved in the reverberator and flash smelting as well as converting operations of Cu-Fe sulfide concentrates to produce matte in the Sar Cheshmeh copper smelter plant, Iran, disposed in the landfill and cooled in air. Due to their relatively high average copper content (about 1.5 wt%), a mineral processing plant based on the flotation process has recently been established to produce thousands of tons of Cu-sulfide concentrate after slag crushing and fine grinding operation. In order to make the flotation process more efficient, more knowledge is required on the form and origin of the copper losses in the slag. To achieve this, mineralogical studies of the slags using optical microscopy, X-ray diffraction (XRD), and scanning electron microscopy (SEM) methods have been carried out. Mineralogical analyses showed the main part of copper losses into the semi- to fully-crystallized magnetite-rich reverberator and flash slags characterized by crystal–glass matrix ratio $\leq 1$ is moderate to coarse particles of Cu-Fe sulfides, i.e., chalcopyrite ($CuFeS_2$) and bornite ($Cu_5FeS_4$), that are mainly chemically entrapped. In contrast, the mechanically entrapped fine- to coarse-grain (from 20 up to 200 μm) spherical-shaped of high-grade matte particles with chalcocite ($Cu_2S$) composition containing droplets or veinlets of metallic copper ($Cu^0$) are the dominant forms of copper losses into the converter slags characterized by crystal–glass matrix ratio $> 1$. From the value recovery point of view, our result show that the fully crystallized slags containing moderate- to coarse-grain copper-bearing particles will result in efficient recovery of a significant amount of entrained copper due to better milling response compared to semi-crystallized ones due to locking the fine- to moderate-grain copper particles in the silicate glassy matrix. Laboratory-scale grinding experiments showed that normal ($\leq 74$ μm) to fine ($\leq 44$ μm) grinding of high- Cu grade slags lead to a significant increase in the liberation degree of copper particles. in contrast, the increase in fine particle fractions ($< 37$ μm) due to re-grinding or ultra-fine grinding of the originally low-Cu grade slags does not lead to the liberation of copper particles, but it will reduce the efficiency of the flotation process. This study suggests that the highest rate of copper recovery of the slag by the flotation process will be obtained at particle size 80% passing 44 μm which has also reached the optimal liberation degree of copper-bearing particles.

**Keywords:** copper slag; mineralogy; copper-bearing particle; particle size; liberation degree

## 1. Introduction

Decreasing high-grade ore deposits with easy processing features and demand for metal products especially base and precious metals (Cu, Zn, Sn, Al, Au, and platinum group elements) are a serious challenge for the metal production industries [1–3]. To overcome this challenge, the recycling of metals from metallurgical wastes (i.e., slag and dust) using conventional methods (such as flotation) and sometimes new ones (such as biotechnology in progress) has long been a part of development plans to increase metal production in the mining industries [4]. Copper slag is one of the most common and important wastes in the mining industry. Since about 80% of the world's copper production

is produced by the pyro-metallurgical process and taking into account leaving 2–3 tons of slag per ton of copper production [5,6], a large volume of slag remains which is estimated at approximately 24.6 million tons of slag for an annual production of 20 million tons of copper in the world and increasing at a rate of 11% per year [7–12]. Stockpiling or disposals of such huge quantities of slag at designated places on the mine site not only requires larger areas but also these slags contain high concentrations of potentially toxic heavy metals which are released into the environment, causing both environmental and space occupation problems [7,10–12]). Copper slag contains large amounts of Cu, Fe, and other valuable metals such as Ni, Co, and Mo. However, these valuable metals are not only a secondary source of metals but also environmental contaminants [10–12]. Therefore, the recovery of valuable metals from the copper slags presents both economic and environmental incentives [10–14]. During the past two decades, attempts have been made to develop methods for metal recovery from copper slags including flotation, magnetic separation, leaching, and roasting [11–14]. Among them, flotation is one of the simplest and most economical methods which efficiently captures metallic Cu and sulfide minerals to provide a copper concentrate, especially for the slag that has been cooled slowly [10,15–18]. Slag processing by flotation is most developed in Japan, Finland, Canada, Philippines, India, USA, Australia, Turkey, Kazakhstan, Russia, and Uzbekistan [15–18].

Copper slag has a Cu content between 0.5 to 3.7 wt% and rarely up to 8 wt% [7–10]. Taking into account the average Cu content of 1.5 wt% [7] and the production of 24.6 million tons per year of slag, it can be estimated that ~370 kt copper is lost into the slag, annually. As a result, copper-producing countries have considered the slag as a secondary mineral reserve in which its Cu grades are higher than primary copper ore deposits [8,10,12]. The knowledge of mineralogy and texture of solidified slag as well as the chemical composition of copper-bearing phases is of particular importance in the industry because depending on the presence and speciation of copper in the slag, the reduction of copper losses into the slag or the slag cleaning process can be suggested [8,10,12].

SarCheshmeh Copper Complex is the first and largest copper producer in Iran, which produces about 250,000 tons of copper per year through pyro-metallurgical and subsequently electrochemical refining operations. During about 40 years of copper production, about 8 million tons of slag with an average Cu content of 1.67 wt%, 40–56 g/t Ag, 1 g/t Au and 0.05 wt% Mo as a metallurgical waste has been disposed of by a way of stockpiling at a place near the refinery plant, and at present, about 750,000 tons per year of slag are added to the slag stockpile (according to data derived from website of National Iranian Copper Industries Company/NICICo). Such a large stockpile of slag with a higher copper content than the primary ore being processed in the SarCheshmeh Copper Complex prompted the NICICo to produce copper concentrate from slag by the flotation process. The flotation plant is now ready to process 1.4 million tons of slag with an average Cu grade of 1.67 wt% to produce 70,000 tons per year of copper sulfide concentrate (unpublished data of NICICo). In order to make the flotation processes more efficient for the recovery of copper from the slag, more knowledge is required on the form and origin of the copper losses in the slag during copper production [6,10,15–18]. Mineralogical characterization of copper in the slags of the SarCheshmeh smelter plant has not been studied to date. The present study deals with the mineralogy of the slag, including the composition, size, and texture of copper-bearing phases, and the type and mode of copper losses in the slag (i.e., mechanical vs. chemical), with the aim of using it in the flotation process to effectively recover copper-bearing phases.

## 2. Materials and Methods

In order to achieve the objectives of the present study, the following methodology was performed.

## 2.1. Sampling and Sample Preparation

Samples for this study were taken from both stock-pilled and operating smelter plants including reverberator and flash smelting furnaces as well as converting furnaces (Figure 1). Totally, 84 bulk samples with a total weight of 420 kg were collected. For sample preparation, each bulk sample was split into two large portions (up to 2500 g for each portion). The first portion of each sample was prepared for polished and polished thin sections to use for mineralogical characterization by optical and electron microscopy. The second portion was crushed in a laboratory-scale jaw crusher to 100% passing 2 mm. About 200 gr of the crushed material was pulverized (80% particles ≤ 44 µm) for chemical and powdered mineralogical characterization (Tables 1–4).

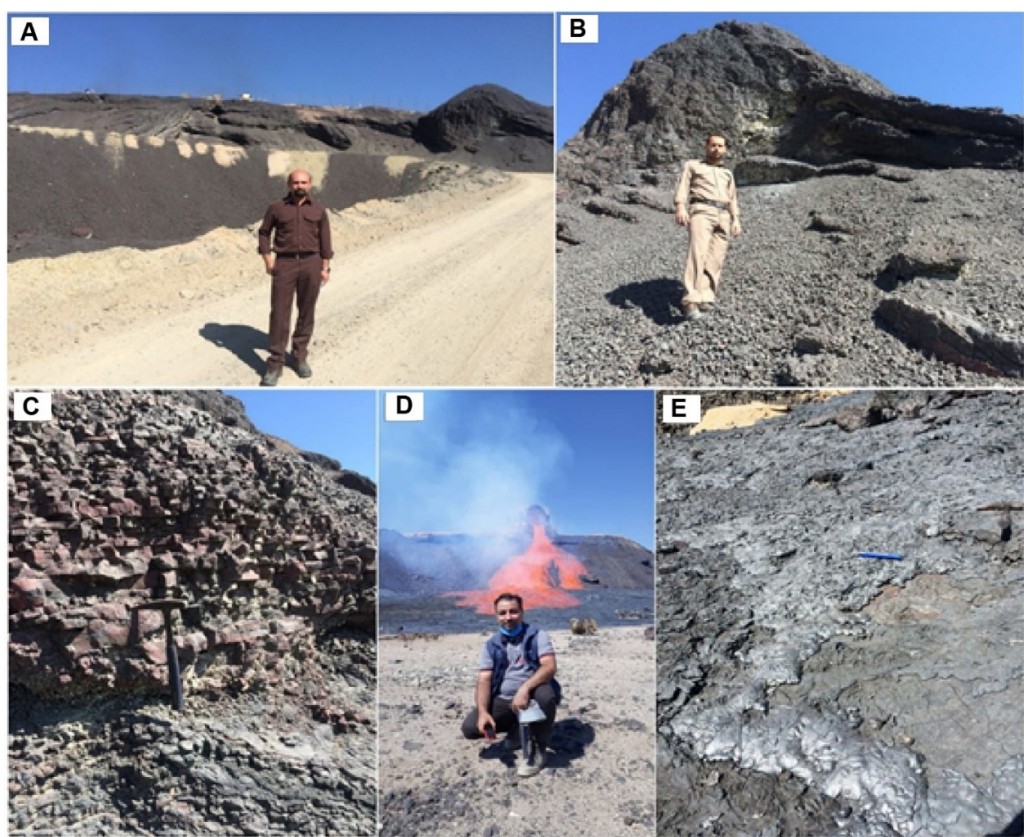

**Figure 1.** Stock-pilled slags in SarCheshmeh Copper Complex; (**A–C**) reverberator furnace slag, (**D**) converter furnace slag, and (**E**) flash furnace slag.

**Table 1.** XRF analytical data of the representative bulk samples of the studied slags. The limit of detections and precision of the analysis is 0.01 wt% and 1.5 wt%, respectively.

| Sample No. | Slag Type | Bulk Chemical Composition (wt%) | | | | | | | | | |
|---|---|---|---|---|---|---|---|---|---|---|---|
| | | $SiO_2$ | $Fe_2O_3$ | $Al_2O_3$ | $SiO_2$ | MgO | $Na_2O$ | $SiO_2$ | $P_2O_5$ | $TiO_2$ | $SiO_2$ | Cu |
| Rb-9009 | Reverberator | 33.55 | Rb-9009 | Reverberator | 33.55 | Rb-9009 | Reverberator | 33.55 | Rb-9009 | Reverberator | 33.55 | Rb-9009 |
| Rb-9023 | Reverberator | 34.02 | Rb-9023 | Reverberator | 34.02 | Rb-9023 | Reverberator | 34.02 | Rb-9023 | Reverberator | 34.02 | Rb-9023 |
| Rb-9025 | Reverberator | 38.75 | Rb-9025 | Reverberator | 38.75 | Rb-9025 | Reverberator | 38.75 | Rb-9025 | Reverberator | 38.75 | Rb-9025 |
| FL-9030 | Flash | 35.35 | FL-9030 | Flash | 35.35 | FL-9030 | Flash | 35.35 | FL-9030 | Flash | 35.35 | FL-9030 |
| FL-9062 | Flash | 48.99 | FL-9062 | Flash | 48.99 | FL-9062 | Flash | 48.99 | FL-9062 | Flash | 48.99 | FL-9062 |
| FL-9031 | Flash | 41.90 | FL-9031 | Flash | 41.90 | FL-9031 | Flash | 41.90 | FL-9031 | Flash | 41.90 | FL-9031 |
| CT-9036 | Converter | 40.67 | CT-9036 | Converter | 40.67 | CT-9036 | Converter | 40.67 | CT-9036 | Converter | 40.67 | CT-9036 |
| CT-9038 | Converter | 32.80 | CT-9038 | Converter | 32.80 | CT-9038 | Converter | 32.80 | CT-9038 | Converter | 32.80 | CT-9038 |
| CT-9046 | Converter | 36.15 | CT-9046 | Converter | 36.15 | CT-9046 | Converter | 36.15 | CT-9046 | Converter | 36.15 | CT-9046 |
| CT-9053 | Converter | 42.41 | CT-9053 | Converter | 42.41 | CT-9053 | Converter | 42.41 | CT-9053 | Converter | 42.41 | CT-9053 |
| CT-9056 | Converter | 41.66 | CT-9056 | Converter | 41.66 | CT-9056 | Converter | 41.66 | CT-9056 | Converter | 41.66 | CT-9056 |

**Table 2.** XRD analytical data of the representative bulk samples of the studied slags. Magnetite content in some samples (indicated by *) determined by SATMAGAN Analyzer. Abbreviation are: Fl = fayalite, Mag = magnetite, Px = pyroxene, Gls = glass/amorphous phase, Ccp = chalcopyrite, Bn = bornite, Cc = chalcocite. The limit of detections is 3 wt% for all phases.

| Sample No. | Slag Type | Bulk Mineralogy (wt%) | | | | | | |
|---|---|---|---|---|---|---|---|---|
| | | Fl. | Mag. | Px. | Fl. | Ccp. | Bn. | Fl. |
| 9009 | Reverberator | 48 | 9009 | Reverberator | 48 | 9009 | Reverberator | 48 |
| 9023 * | Reverberator | 49 | 9023 * | Reverberator | 49 | 9023 * | Reverberator | 49 |
| 9025 * | Reverberator | 48 | 9025 * | Reverberator | 48 | 9025 * | Reverberator | 48 |
| 9030 * | Flash | 45 | 9030 * | Flash | 45 | 9030 * | Flash | 45 |
| 9062 * | Flash | 38 | 9062 * | Flash | 38 | 9062 * | Flash | 38 |
| 9031 * | Flash | 44 | 9031 * | Flash | 44 | 9031 * | Flash | 44 |
| 9036 * | Converter | 50 | 9036 * | Converter | 50 | 9036 * | Converter | 50 |
| 9038 | Converter | 46 | 9038 | Converter | 46 | 9038 | Converter | 46 |
| 9046 | Converter | 47 | 9046 | Converter | 47 | 9046 | Converter | 47 |
| 9053 * | Converter | 44 | 9053 * | Converter | 44 | 9053 * | Converter | 44 |
| 9056 | Converter | 46 | 9056 | Converter | 46 | 9056 | Converter | 46 |
| 9017 * | Reverberator | 49 | 9017 * | Reverberator | 49 | 9017 * | Reverberator | 49 |
| 9024 * | Reverberator | 44 | 9024 * | Reverberator | 44 | 9024 * | Reverberator | 44 |
| 9028 * | Reverberator | 46 | 9028 * | Reverberator | 46 | 9028 * | Reverberator | 46 |
| 9060 * | Flash | 49 | 9060 * | Flash | 49 | 9060 * | Flash | 49 |
| 9074 * | Flash | 48 | 9074 * | Flash | 48 | 9074 * | Flash | 48 |
| 9034 | Converter | 42 | 9034 | Converter | 42 | 9034 | Converter | 42 |
| 9062 * | Converter | 50 | 9062 * | Converter | 50 | 9062 * | Converter | 50 |
| 9035 | Converter | 44 | 9035 | Converter | 44 | 9035 | Converter | 44 |
| 9040 | Converter | 41 | 9040 | Converter | 41 | 9040 | Converter | 41 |
| 9041 | Converter | 43 | 9041 | Converter | 43 | 9041 | Converter | 43 |
| 9054 * | Converter | 50 | 9054 * | Converter | 50 | 9054 * | Converter | 50 |

**Table 3.** Cu content (wt%) in the sized fractions related to the first mode grinding of the bulk slag samples. The limit of detections and precision of the analysis using XRF to measure the Cu content is 0.01 wt% and 1.5 wt%, respectively.

| Sample No. | Slag Type | Cu Content (wt%) in the Sized Fractions | | | | | |
|---|---|---|---|---|---|---|---|
| | | ≥74 | 63 | 53 | ≥74 | 37 | <37 |
| Rb-9009 | Reverberator | 3.05 | Rb-9009 | Reverberator | 3.05 | Rb-9009 | Reverberator |
| Rb-9023 | Reverberator | 2.75 | Rb-9023 | Reverberator | 2.75 | Rb-9023 | Reverberator |
| Rb-9025 | Reverberator | 0.48 | Rb-9025 | Reverberator | 0.48 | Rb-9025 | Reverberator |
| Rb-9001 | Reverberator | 0.24 | Rb-9001 | Reverberator | 0.24 | Rb-9001 | Reverberator |
| FL-9030 | Flash | 6.88 | FL-9030 | Flash | 6.88 | FL-9030 | Flash |
| FL-9032 | Flash | 0.51 | FL-9032 | Flash | 0.51 | FL-9032 | Flash |
| FL-9062 | Flash | 0.98 | FL-9062 | Flash | 0.98 | FL-9062 | Flash |
| CT-9046 | Converter | 1.21 | CT-9046 | Converter | 1.21 | CT-9046 | Converter |
| CT-9038 | Converter | 9.15 | CT-9038 | Converter | 9.15 | CT-9038 | Converter |
| CT-9034 | Converter | 8.58 | CT-9034 | Converter | 8.58 | CT-9034 | Converter |

To reveal the role of the grinding rate of the slag in the liberation of copper-bearing particles, high-Cu and low-Cu content samples from all types of the studied slags (i.e., reverberator, flash, and converter slags) were ground in 2 different modes. One mode includes grinding the crushed slag samples (particle size < 2 mm) under the same condition of milling over 20 min to reduce their particle size to <74 μm, then particle sizing the ground samples into the 5 size fractions including >63 μm, 53 μm, 44 μm, 37 μm, and <37 μm (Table 3). Another mode involves grinding of the crushed slag samples (particle size < 2 mm over four separated time periods including 10, 15, 20, and 25 min, respectively, corresponding to reaching 50 vol%, 60 vol%, 70 vol%, and 80 vol% of the particle size of the ground samples to ≤44 μm (Table 4). To achieve this, a large portion of each

crushed slag (~1200 gr) was weighed and added to a laboratory-scale ball mill loaded with 20 kg grinding steel balls. Then, 600 mL of water was added to the charge in order to achieve 66% solids by mass slurry. After grinding the samples in two different modes mentioned above, the mill discharge was washed out into the bucket over a wire mesh. The product slurry was then dewatered using a filter press, and dried in the oven set at a temperature of 40 °C for one day. Screening the ground and dried slag samples using a shaking sieve led to production of fine-sized fractions of samples with particle sizes ranging from ≥63 to ≤37 μm for the first grinding mode, and four fractions with different percentages of particle size ≤ 44 μm for the second mode of slag grinding (Tables 3 and 4). Then, all fractions were analyzed by XRF to measure their Cu content (wt%), and optical microscopic counting method was used to obtain the liberation degree of copper-bearing particles (Tables 3 and 4). In this method, free $(n_1)$ and locked copper-bearing particles $(n_2)$ were counted using a reflected light microscope and the liberation degree of copper-bearing particles was calculated from the following formula: Ld% = $(n_1)/(n_1 + n_2) \times 100$ (Table 4). In order to determine the density of the slag samples, a 20 gr of fine-sized slag sample (particle size ≤ 74 μm) was weighted. Then, the weight of the 100 mL cylinder in the dry state $(M_1)$ was measured. By adding the specified amount of water into the cylinder, the weight of the cylinder and water $(M_4)$ was determined and recorded. Following that, 5 g out of the 20 g of fine-sized slag sample (solid) was weighed and added to the cylinder $(M_2)$, then followed by adding more water gradually to the cylinder. In the last stage, the weight of the cylinder, water, and solid $(M_3)$ was measured and recorded by the analytical balance. The weights were measured and recorded in three steps, and these procedures were repeated twice for each sample (Totally 86 samples). Finally, the solid density $(g/cm^3)$ was calculated using the following formula: $D_S = M/V*D_W$ after obtaining the solid weight by $M = M_2 + M_1$, solid volume by $V = (M_4 - M_1) - (M_3 - M_2)$, and water density by $D_W = M/100$.

**Table 4.** Cu content (wt%) in the representative combined slag samples and Cu content (wt%) as well as the liberation degree of copper-bearing particles in the sized fractions related to the second mode grinding of the representative combined slag samples. The limit of detections and precision of the analysis using XRF to measure the Cu content is 0.01 wt% and 1.5 wt%, respectively.

| Representative Combined Slag Samples | Slag Type | Cu (wt%) | Sized Fractions of Ground Slag | Particle Size Distribution | Cu (wt%) | Liberation Degree of Copper-Bearing Particles (Ld.%) |
|---|---|---|---|---|---|---|
| HRb-RF | High-Cu grade reverberator slag | 0.90 | HRb50-RF | 50% particle size ≤ 44 μm | 0.95 | 15 |
| | | | HRb60-RF | 60% particle size ≤ 44 μm | 0.97 | 23 |
| | | | HRb70-RF | 70% particle size ≤ 44 μm | 1.05 | 36 |
| | | | HRb80-RF | 80% particle size ≤ 44 μm | 1.12 | 42 |
| LRb-RF | Low-Cu grade reverberator slag | 0.75 | LRb50-RF | 50% particle size ≤ 44 μm | 0.68 | 12 |
| | | | LRb60-RF | 60% particle size ≤ 44 μm | 0.70 | 26 |
| | | | LRb70-RF | 70% particle size ≤ 44 μm | 0.70 | 29 |
| | | | LRb80-RF | 80% particle size ≤ 44 μm | 0.75 | 35 |
| CT-RF | Converter slag | 1.14 | CT50-RF | 50% particle size ≤ 44 μm | 1.51 | 23 |
| | | | CT60-RF | 60% particle size ≤ 44 μm | 1.65 | 14 |
| | | | CT70-RF | 70% particle size ≤ 44 μm | 1.67 | 24 |
| | | | CT80-RF | 80% particle size ≤ 44 μm | 1.73 | 24 |
| FL-RF | Flash slag | 1.69 | FL50-RF | 50% particle size ≤ 44 μm | 1.10 | 24 |
| | | | FL60-RF | 60% particle size ≤ 44 μm | 1.12 | 22 |
| | | | FL70-RF | 70% particle size ≤ 44 μm | 1.18 | 38 |
| | | | FL80-RF | 80% particle size ≤ 44 μm | 1.23 | 45 |

### 2.2. Analytical Methods

To obtain the chemical composition of the prepared slag samples, small, pressed powder mounts of the samples were used for analysis by X-ray fluorescence spectrometry (XRF). Intensity measurements were carried out using a Philips PW 1480 wavelength-dispersive X-ray spectrometer (2.4 kW 400 mA) with the following experimental conditions: X-ray tube containing a Rh anode, LiF (200), and Ge analyzer crystals, a fine collimator (150 μm), flow and scintillation detectors and the appropriate spectrometer mask (35 mm diameter). A counting time of 100s was used in all cases.

The mineralogical characterization of the bulk, representative, and ground slag samples was carried out by X-ray diffraction (XRD), magnetic analyzer, and optical and scanning electron microscope (SEM) instruments. Bulk mineralogy of the slag samples ($\leq$74 μm) was performed by X-ray diffraction analysis (XRD: Philips Pw 1800, Netherland) using a mono-chromatic CuK radiation operated at 40 kV and 30 mA. Data were collected at 2θ from ~5° to 70°.

In addition to XRD, we used Satmagan S135 Magnetic Analyzer (Rapiscan Systems, Torrance, CA, USA) to determine the magnetite content in some samples of the studied slags based on measurement of the magnetic moment (m) after the magnetic component in the sample (V) has been magnetized for saturation (Msat). The magnetic moment is determined by measuring the force acting on the sample in a non-homogeneous magnetic field (F) having a vertical gradient of (dH/dz), and comparing it with the gravitational force (G) acting on the sample: F/G = [m(dH/dz)]/gmt = [VMsat(dH/dz)]/gmt = [Msat(dH/dz)]/gp × mm/mt, in which g = gravitational constant (9.8 m/s$^2$), mt = total mass of the sample, mm = mass of the magnetic component in the sample, and p = density of the magnetic component. The percentage of the magnetic components in the samples is obtained by 100 × msat/mt = 100 × (pg)/[msat(dH/dz)] × F/G. Measuring the ratio of the magnetic to the gravitational force (F/G) and multiplying this by a coefficient give the percentage of the magnetite content in the samples.

Optical microscopy was performed by using a reflected–transmitted microscope (Olympus, BX50, Japan) under different objective lens magnification (4×–40×). Micro-imaging and micro-analysis of the slag samples were performed by using an SEM (CamScan Electron Optics Ltd., Water beach, Cambridge, UK) equipped with an energy dispersive X-ray spectrometer (EDS; Oxford Inca, Oxon, UK), and back-scattered (BSE) as well as secondary electron detectors. Semi-quantitative chemical analyses were carried out by EDS using an accelerating voltage of 15 kV and a beam current of 20 mA. Representative BSE images of the slag samples were analyzed using ImageJ software, a Java-based image processing program.

All quantitative analytical data obtained by XRF, XRD, and MAGNETIC ANALYZER of the selected bulk samples and those that were subjected to the grinding tests are presented in Tables 1–4.

### 3. Results

#### 3.1. Physical Properties

Slag samples of reverberator furnaces were mostly heavy, hard, and black in color, and some samples were relatively porous compared to the less porous samples that had a dense texture (Figure 2A–C). Compared to the reverberator slag, the flash and converter slags have a metallic luster, dark to light gray in color but have less porosity (Figure 2D,F). Copper-bearing phases (CBP) were more or less visible in the form of millimeter-sized shiny droplets and veinlets as well as rounded grains on the fresh or polished surface of all types of the studied slags (Figure 2). Slag density varied between 3.5 to 4.8 g/cm$^3$, and there is no obvious difference between slag types (Figure 3).

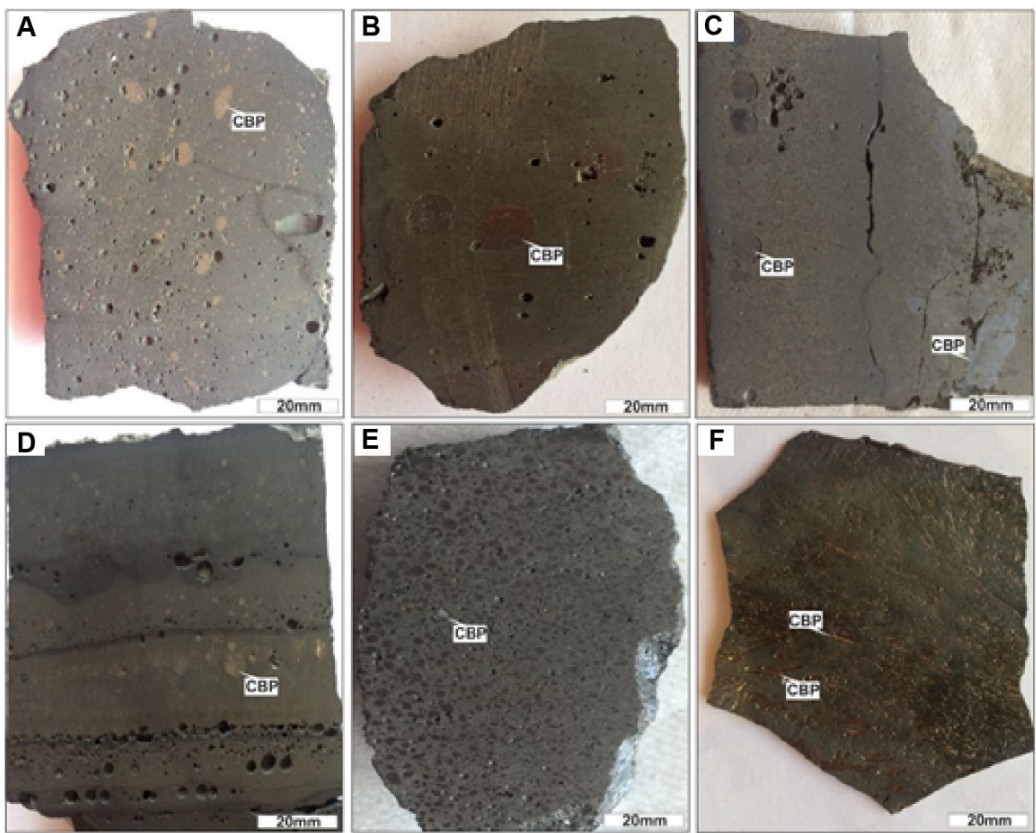

**Figure 2.** Photographs showing the physical properties of the studied reverberator (**A**–**C**), flash (**D**,**E**), and converter slags (**F**). The reverberator furnace slag has a porous surface and is oxidized. Additionally, flash and converter furnaces slags have a metallic luster. In the polished slabs of slags, the copper-bearing phases (i.e., CBP) as shiny droplets and veinlets are observed.

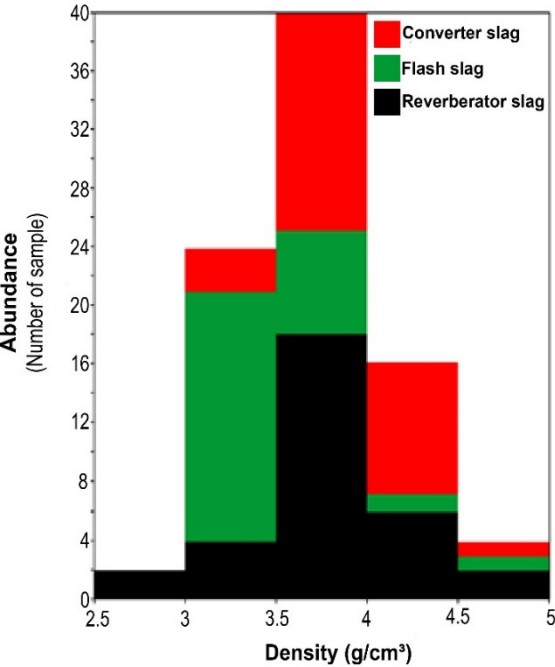

**Figure 3.** Comparison of density in the studied slag samples.

### 3.2. The Bulk Chemistry

The results of XRF analysis of the representative bulk samples (Table 1) showed that Fe ($Fe_2O_3$) with a concentration of 34 to 58 wt% and Si ($SiO_2$) with a concentration of 22 to 50 wt% are the main chemical components. (Figure 4). The next abundant elements are Al ($Al_2O_3$) and Ca (CaO), each of which reaches up to 10 wt% (Figure 4). The Cu concentration of the studied slags varied from 0.78 to 18.75 wt% (Figure 4). The reverberator slags show lower Cu content (0.34–5.75 wt% with an average of 1.77 wt%) compared to the flash (up to 18.75 wt% with an average of 2.44 wt%) and the converter slags (1.21–18.65 wt% with an average of 4.65 wt%). A relatively positive relationship between Cu content of the slags and their density was observed, especially in the flash and converter slag samples (Figure 5).

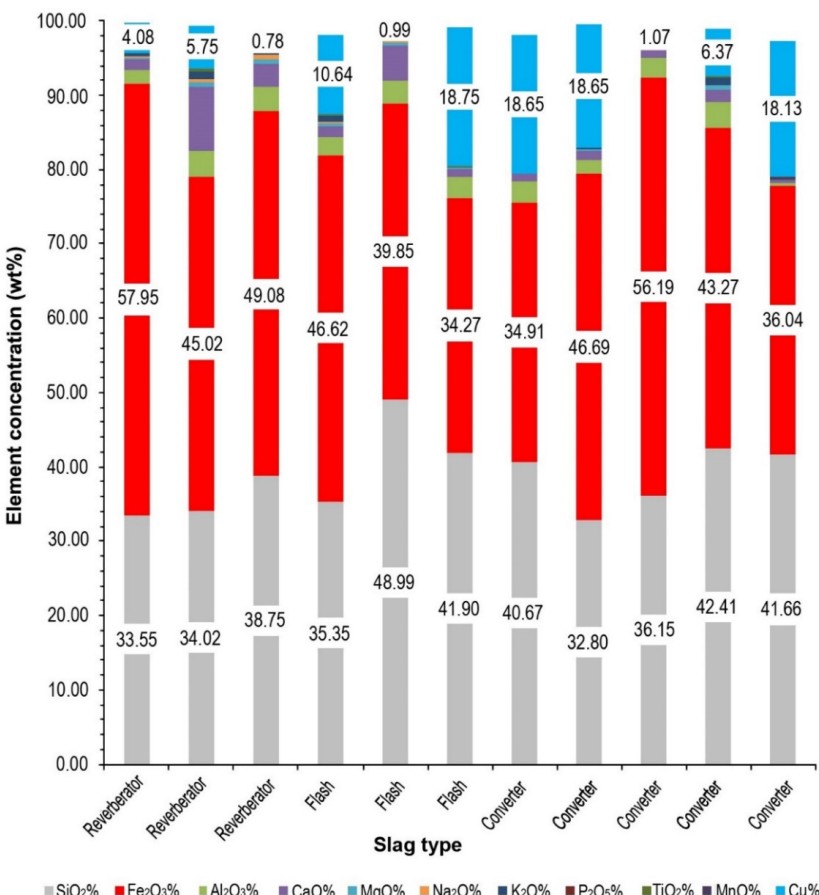

**Figure 4.** Bulk chemistry of the studied slags.

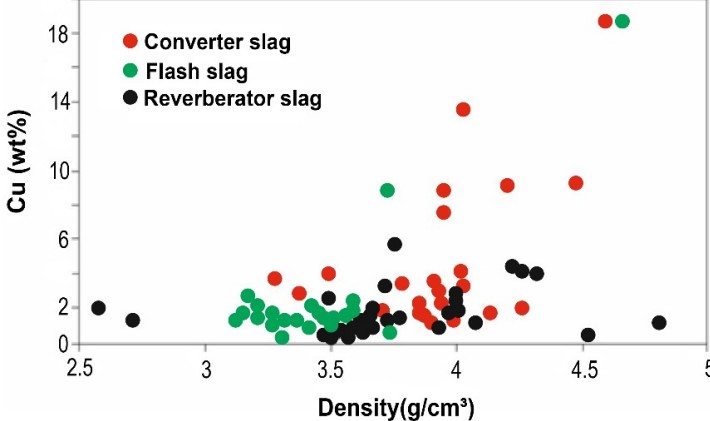

**Figure 5.** Relationship between Cu content (wt%) and density (g/cm$^3$) in the studied slags.

### *3.3. Mineralogical Composition*

### 3.3.1. Bulk Mineralogy

XRD analysis of a number of samples shows fayalite ($Fe_2SiO_4$) is the major crystalline phase up to 50 wt% in the studied slags (Figure 6). In addition to fayalite, magnetite ($Fe_3O_4$) was the second abundant crystalline phase in the slags (Table 2 and Figures 6 and 7). The magnetite content in the studied slags varied from 4 to 28 wt% with the maximum values in the converter and reverberator slags (Table 2 and Figure 7). Flash furnaces slags showed the lowest magnetite content (between 4 and 7 wt%). In samples with Cu content of more than 3 wt%, XRD analysis revealed the presence of Cu sulfides such as bornite ($Cu_5FeS_4$), chalcocite ($Cu_2S$), and chalcopyrite ($CuFeS_2$) along with the main crystalline phases (Table 2 and Figure 6).

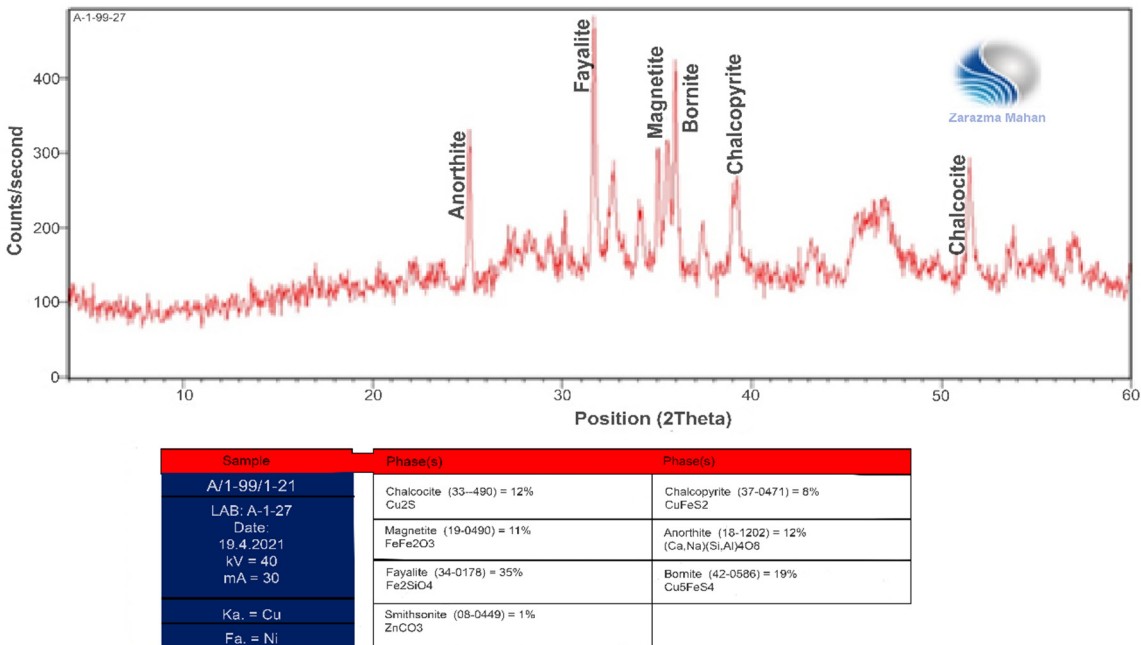

**Figure 6.** XRD graph of a Cu-rich converter slag (9.35 wt% Cu).

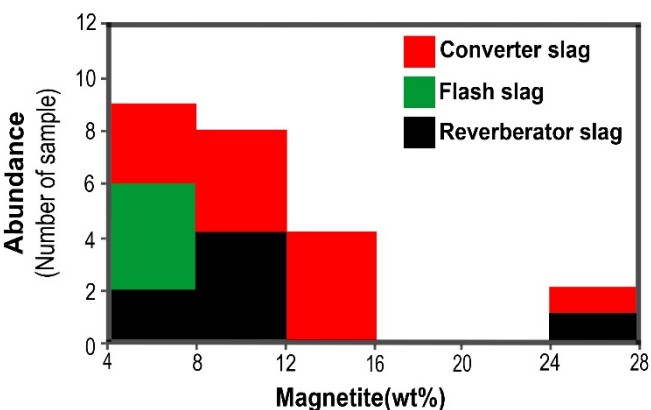

**Figure 7.** Magnetite content in the studied slags.

### 3.3.2. Mineralography

Mineralography using a reflected light microscope showed that the reverberator and flash smelting furnace slag samples are characterized by a range of textures from typical spinifex, to semi- and/or fully crystalline textures compared to hypocrystalline texture in converter smelting furnace slag samples (Figures 8–12). All slag samples are made almost entirely of fayalite (50–70 vol%) and magnetite (15–35 vol%) crystals, with different

amounts of interstitial silicate glass phase from the lowest amount in flash and converter slag samples (up to 10 vol%) to the highest amount in reverberator ones (from 5 up to 40 vol%) (Figures 8–12).

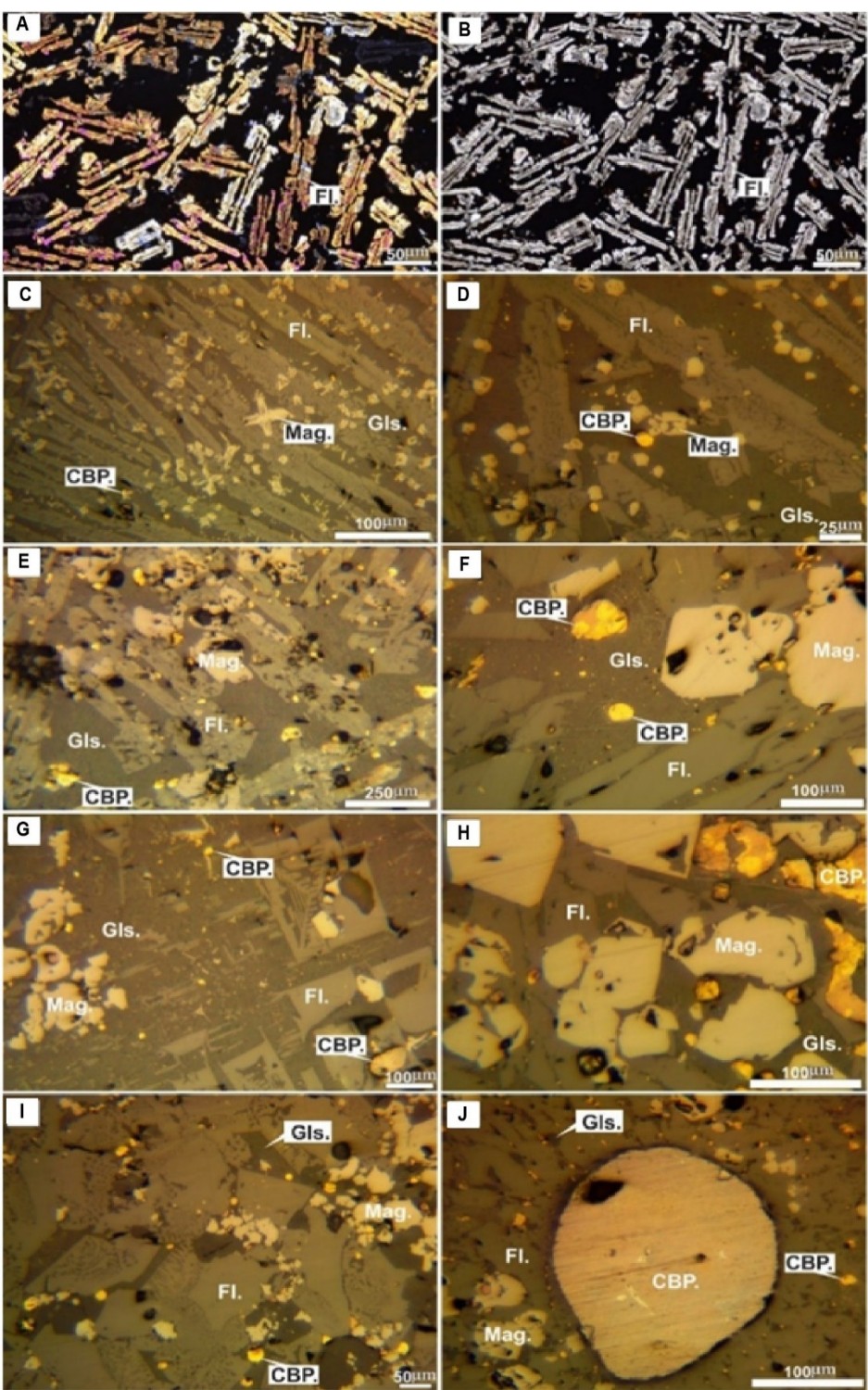

**Figure 8.** Microphotographs showing mode occurrence of fayalite phase in the reverberator (**A,B,D–F,H,I** microphotographs), flash (**C,G**) microphotographs) and converter ((**J**) microphotograph) slags. (**A,B**) microphotographs were taken in transmitted light mode, and others were taken in reflected light mode. See the related text for more details. CBP = copper-bearing phases, Fl = fayalite, Mag = magnetite, Gls = glass.

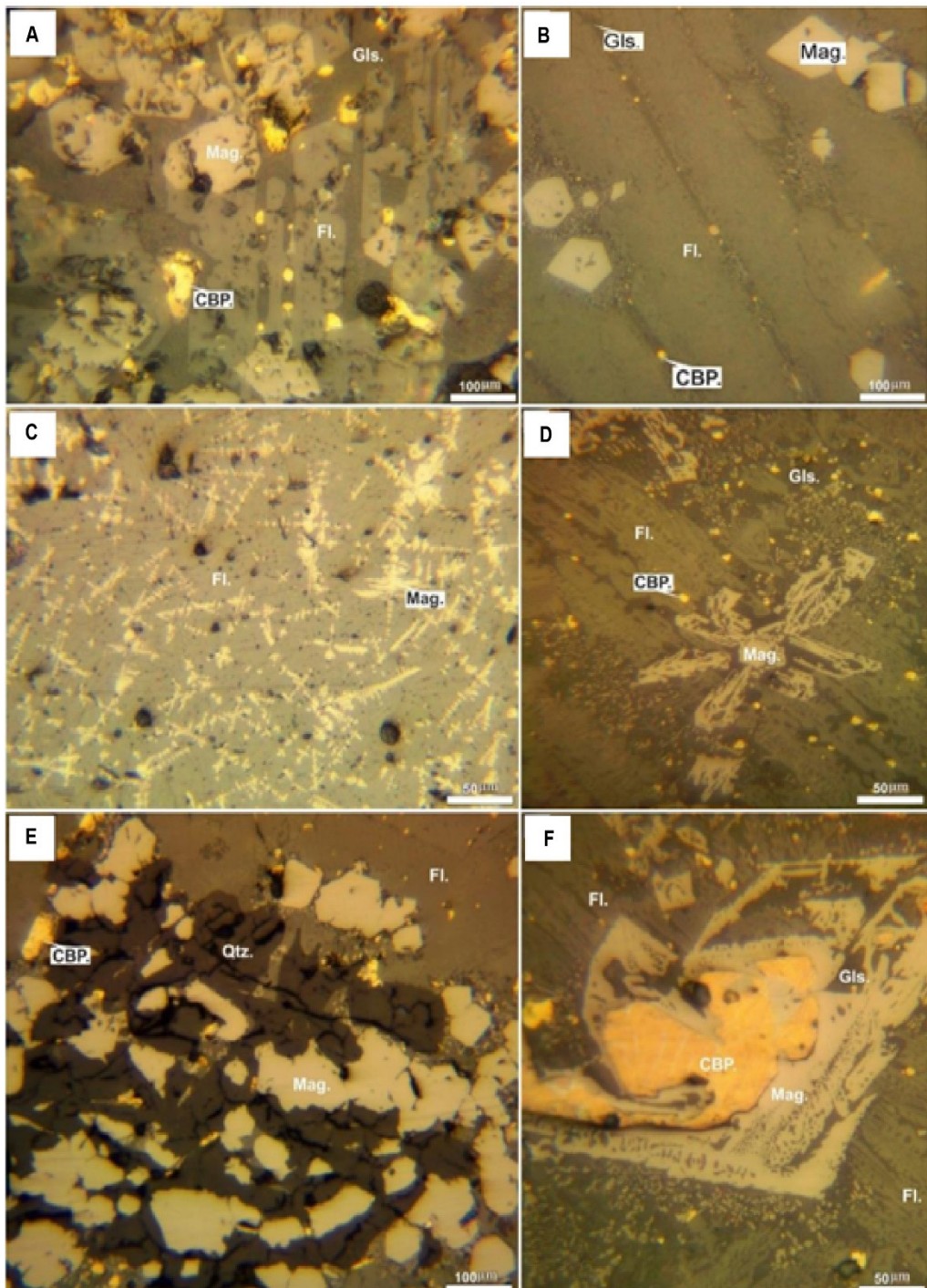

**Figure 9.** Microphotographs showing mode occurrence of magnetite in the reverberator (**A**,**B**,**D**) microphotographs), flash ((**F**) microphotograph), and converter slags ((**C**,**E**) microphotographs). See the related text for more details. CBP: copper-bearing phases, Fl = fayalite, Mag = magnetite, Gls = glass, Qtz = quartz.

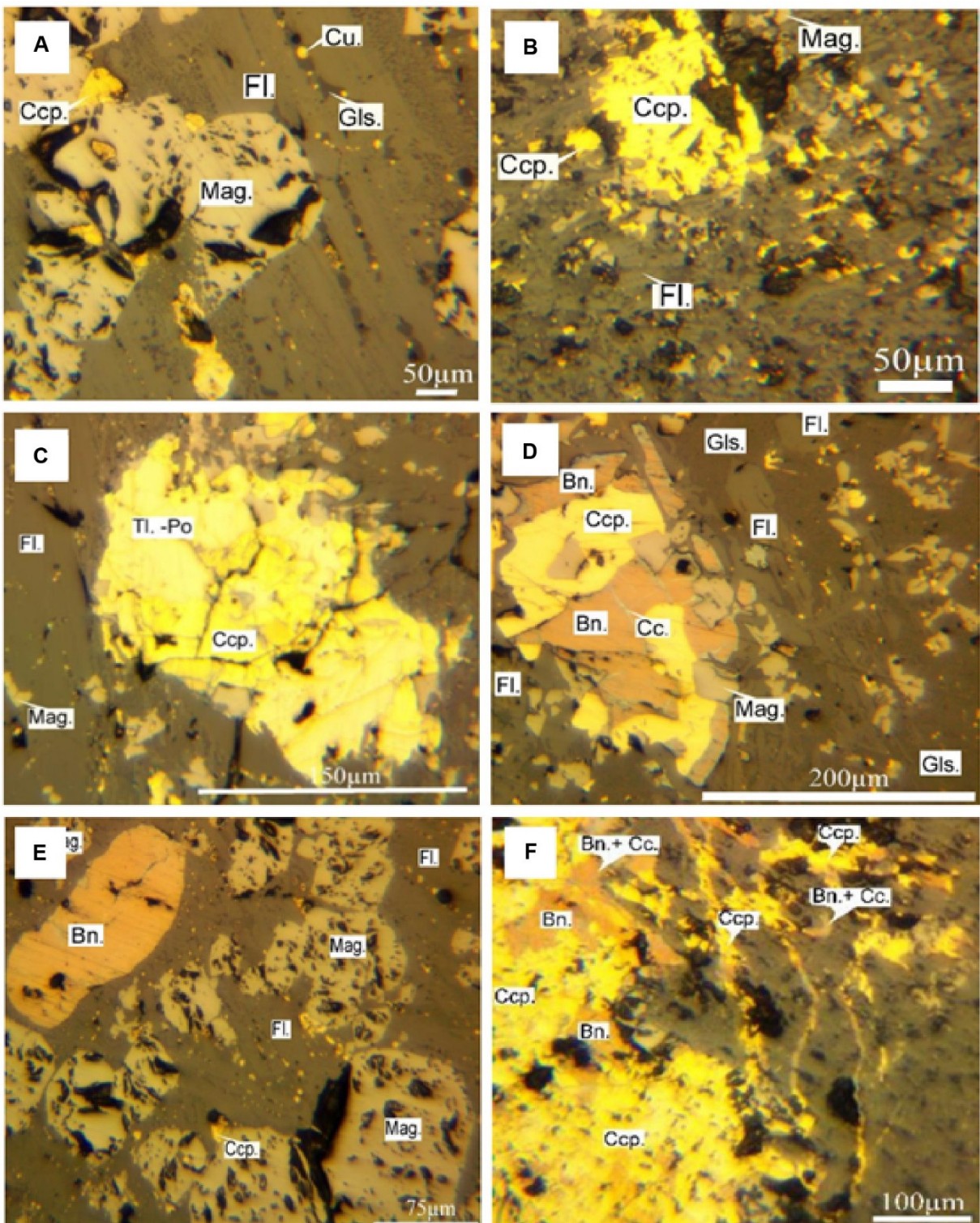

**Figure 10.** Microphotographs showing mode occurrence of copper-bearing phases in reverberator slag samples. See the related text for more details. Bn = bornite, Cc = chalcocite, Ccp = chalcopyrite, Po = pyrrhotite, Tl = troilite, Cu = metallic copper, Fl = fayalite, Mag = magnetite, Gls = glass.

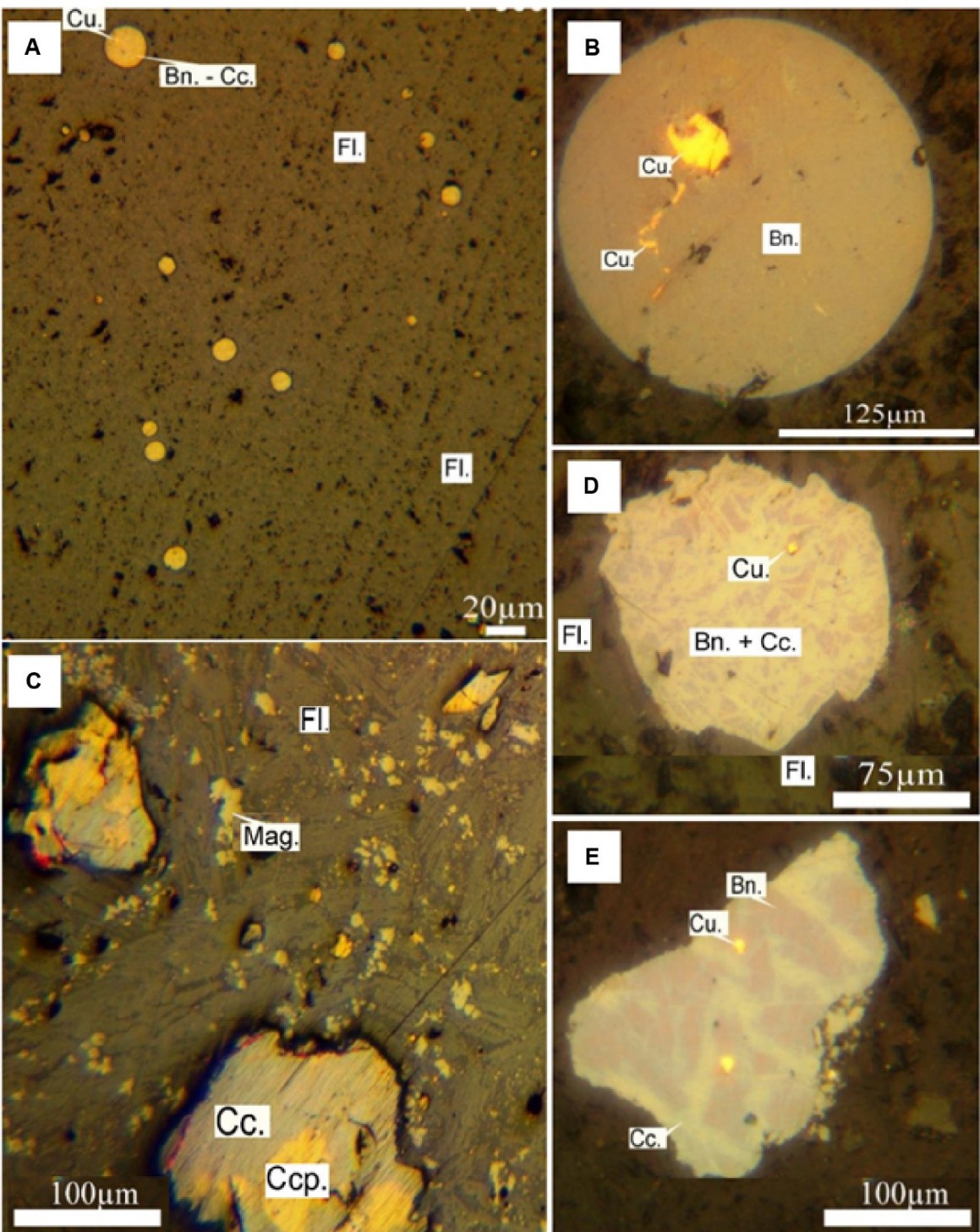

**Figure 11.** Microphotographs showing mode occurrence of copper-bearing phases in flash slag samples. See the related text for more details. Bn = bornite, Cc = chalcocite, Ccp = chalcopyrite, Cu = metallic copper, Fl = fayalite, Mag = magnetite, Gls = glass.

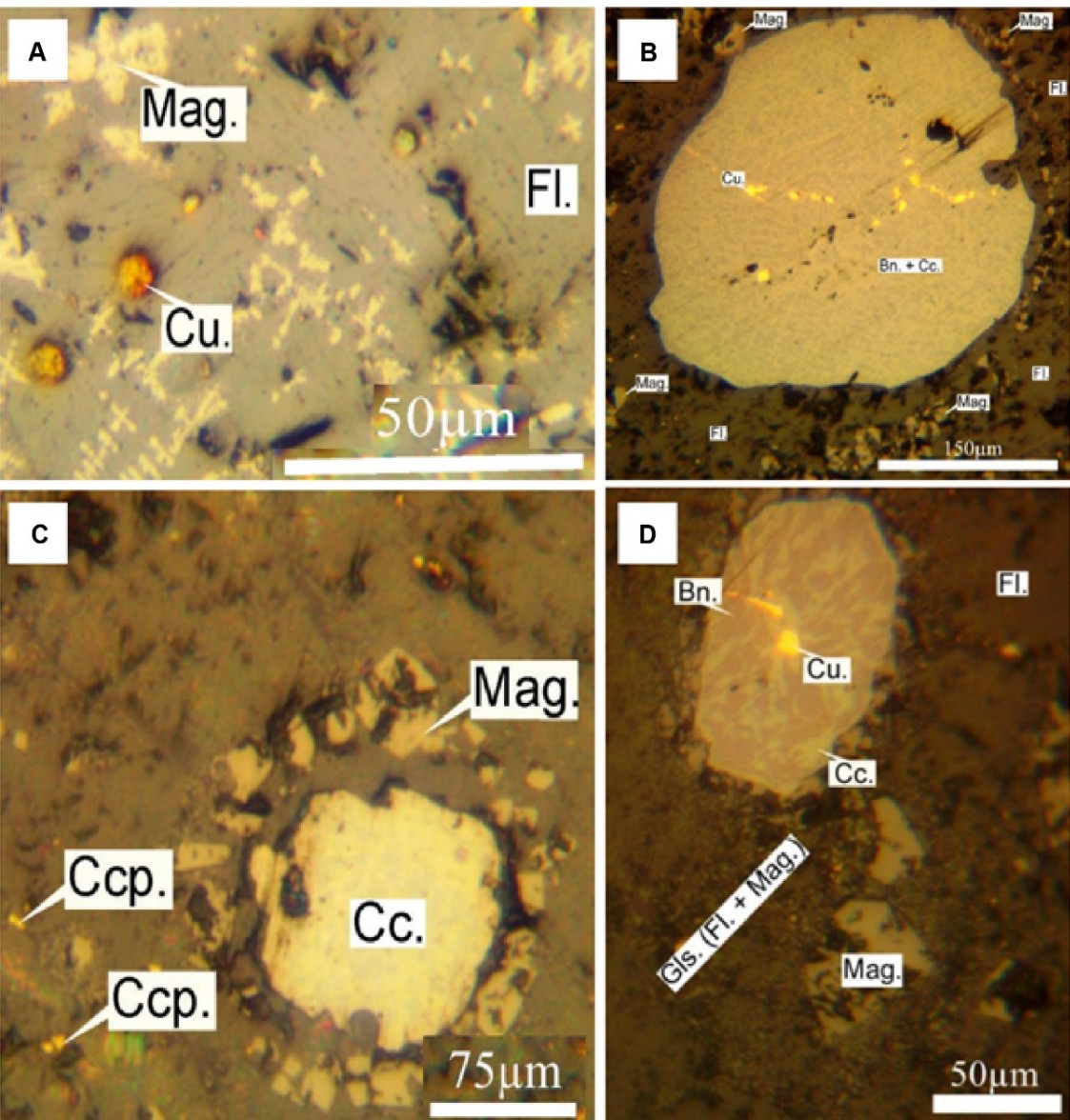

**Figure 12.** Microphotographs showing mode occurrence of copper-bearing phases in converter slag samples. See the related text for more details. Bn = bornite, Cc = chalcocite, Ccp = chalcopyrite, Cu = metallic copper, Fl = fayalite, Mag = magnetite, Gls = glass.

Fayalite as the predominant crystalline phases commonly show textures in the form of elongated, large- to medium-sized columnar, lath- or spear-like subparallel or radiating growths crystals (up to several hundreds of micrometers in length) and also irregularly shaped crystals (Figure 8). Parallel and semi-parallel elongated and/or irregularly shaped crystals of fayalite completely enclose the magnetite and copper-bearing phases which are scattered throughout the silicate glassy matrix (Figure 8C–J).

The most abundant phase that accompanies fayalite in all studied slags is magnetite. The reverberator and converter slags contain the largest proportions of magnetite compared to flash ones. Texturally, magnetite mainly occurred as euhedral to subhedral grains (square-like and round shape crystals) up to 250 μm diameter, mostly scattered between fayalite crystals, within the silicate glassy matrix (Figures 8C–J and 9A,B). In some cases of flash and converter slag samples, magnetite shows transitional textures to hematite in the form of tower-like and spindle-like (Figure 9C,D). Quartz is one of the phases accompanying magnetite and fayalite in a small number of reverberator slag samples (Figure 9E). Quartz

was not observed as a euhedral and primary phase, but it mostly appeared as a phase replacing magnetite (Figure 9E). Additionally, in some cases, the copper phases were surrounded by magnetite (Figure 9F).

The most common copper-bearing phases in the studied slags were Cu $\pm$ Fe sulfides, with optical properties similar to the main copper ore minerals such as chalcopyrite, bornite, and chalcocite. According to the type of smelting process, copper sulfide phases show different compositions, sizes, and textures. In the reverberator slag samples, the most abundant copper sulfide phases were chalcopyrite and bornite which were present in the form of both individual and mineral pairs (Figure 10). In the case of individual form, round-shaped and fine-size grains of chalcopyrite phase (between 50 and 100 μm) were observed as disseminated particles within the interstitial silicate glass matrix (Figure 10A), whereas irregularly shaped medium-to coarse-size grains of chalcopyrite phase (up to ~250 μm) were present in the semi- to fully crystalline matrix of the slag along with fayalite and magnetite (Figure 10A,B). In very rare cases, coarse-size grains of chalcopyrite phase contain troilite (FeS) and pyrrhotite [Fe$(1 - x)$S$(x = 0 - 0.17)$] as intergrowth mineral assemblage were observed (Figure 10C). The association of chalcopyrite and bornite as a mineral pair was one of the most common occurrences of Cu-Fe sulfide phases in the reverberator slag samples (Figure 10D–F). In this case, the chalcopyrite–bornite mineral pair has mostly transformation texture showing partial replacement of chalcopyrite by bornite (Figure 10D–F). Additionally, the coexistence of chalcopyrite with bornite and magnetite (i.e., simultaneous crystallization) was also observed in rare cases (Figure 9D). Large elongated-, ovoid-, and/or teardrop-shaped grains of bornite phase (ca. 250 μm in size) were found in some samples of the reverberator slag (Figure 10E). Chalcopyrite $\pm$ bornite veinlets were other but rare forms of copper sulfide phases in the studied slags that cut the fully crystalline matrix (Figure 10F). The metallic copper phase in the form of tiny matte particles (<10 μm) disseminated throughout the interstitial silicate glass phase, mostly occupying the gaps between the large elongated skeletal crystals of fayalite, was another form of copper-bearing phases in the reverberator slag samples (Figure 10A). In contrast, flash and converter slag samples a significant difference in the form and composition of copper-bearing phases. The matte particles as the main copper-bearing phases in the flash and converter slags displayed a greater diversity in shape, including spherical, elongate, ovoid, and teardrop shapes, and vary widely in size (from ~10 μm to up to 200 μm) and circularity and roundness, even within the same slag sample (Figures 11 and 12). In terms of mineralogical composition and internal texture, the matte grains are mostly chalcocite and bornite in composition, which mainly show the solid solution and/or symplectic intergrowth textures containing inclusions and veinlets of the metallic copper phase (Figures 11A–E and 12A–D). Chalcopyrite, a less common copper-bearing phase in the flash and converter slags, in the cases that were observed showed partial transformation to chalcocite (Figure 11C). Due to leaving molten slag in landfills and relatively slow cooling in the air (air-cooled mode), the glass phase has formed in many cases as the interstitial silicate glass containing micron-sized inclusions of metallic copper and Cu-sulfides phases (Figures 8C,H–J and 9B). In cases where the slag has cooled quickly, the glass phase has formed with a larger volume, and the main phases (i.e., fayalite and magnetite) have crystallized in a silicate glassy matrix (Figure 8D,G). According to SEM analysis, the glass phase has a mineralogical composition like clinopyroxene (hedenbergite–diopside) and Ca-rich plagioclase mineral assemblage.

### 3.3.3. Mineral Chemistry

Using SEM including both BSE imaging and EDS elemental determinations of many point analyses (n = 43 points) in this study revealed a wide range of Cu, Fe, and S concentrations in the selected copper-bearing phases (Figures 13–18). Six groups of copper-bearing phases were recognized as followed:

(a)  Metallic copper group: copper-bearing phases with Cu concentration $\geq$ 90 wt% and also minor S and Fe impurities

(b) Chalcocite ($Cu_2S$) group: copper-bearing phases with a chemical composition similar to the theoretical composition of chalcocite with a concentration range of Cu between 70 and 80 wt%, and 20–30 wt% S.

(c) Idaite ($Cu_5FeS_6$; 56.14 wt% Cu, 33.99 wt% S, 9.87 wt% Fe)–Bornite ($Cu_5FeS_4$; 63.31 wt% Cu, 25.56 wt% S, 11.13 wt% Fe) group: copper-bearing phases with a chemical composition similar to the theoretical composition of idaite and bornite with Cu concentrations between 40 and 70 wt% and a significant amount of S and a minor amount of Fe.

(d) Chalcopyrite ($CuFeS_2$; 34.63 wt% Cu, 34.94 wt% S, 30.43 wt% Fe)–Cubanite ($CuFe_2S_3$; 23.41 wt% Cu, 35.44 wt% S, 41.15 wt% Fe) group: copper-bearing phases with a chemical composition similar to the theoretical composition of chalcopyrite and cubanite with Cu concentrations between 20 and 40 wt% and a significant amount of S and Fe.

(e) Copper–Iron–Sulfur isomorphic compound containing the same amount of Cu, Fe, and S.

(f) Copper-bearing Fe-S isomorphic phases (less than 10 wt% Cu) such as copper-bearing pyrrhotite $Fe_{(1-x)}S_{(x=0-0.17)}$.

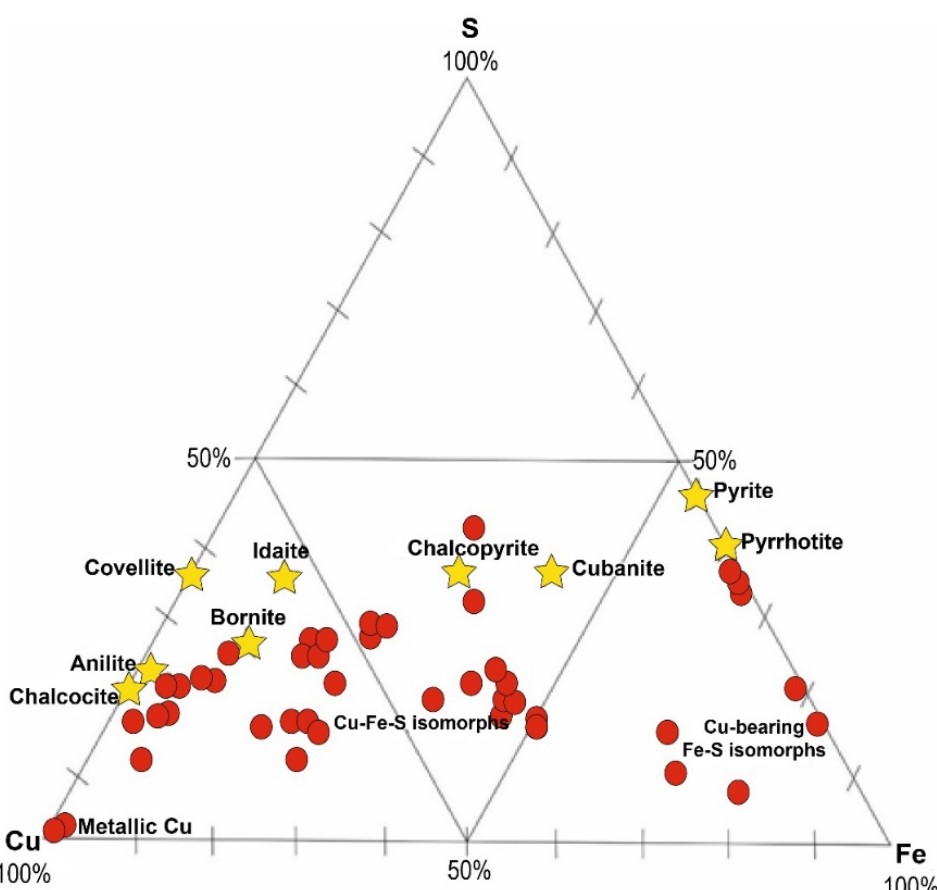

**Figure 13.** Mineralogical composition of copper-bearing phases in the studied slags based on SEM/EDS elemental determinations of point analyses (n = 43 points). The star symbol represents the theoretical chemical composition of Cu±Fe sulfide minerals.

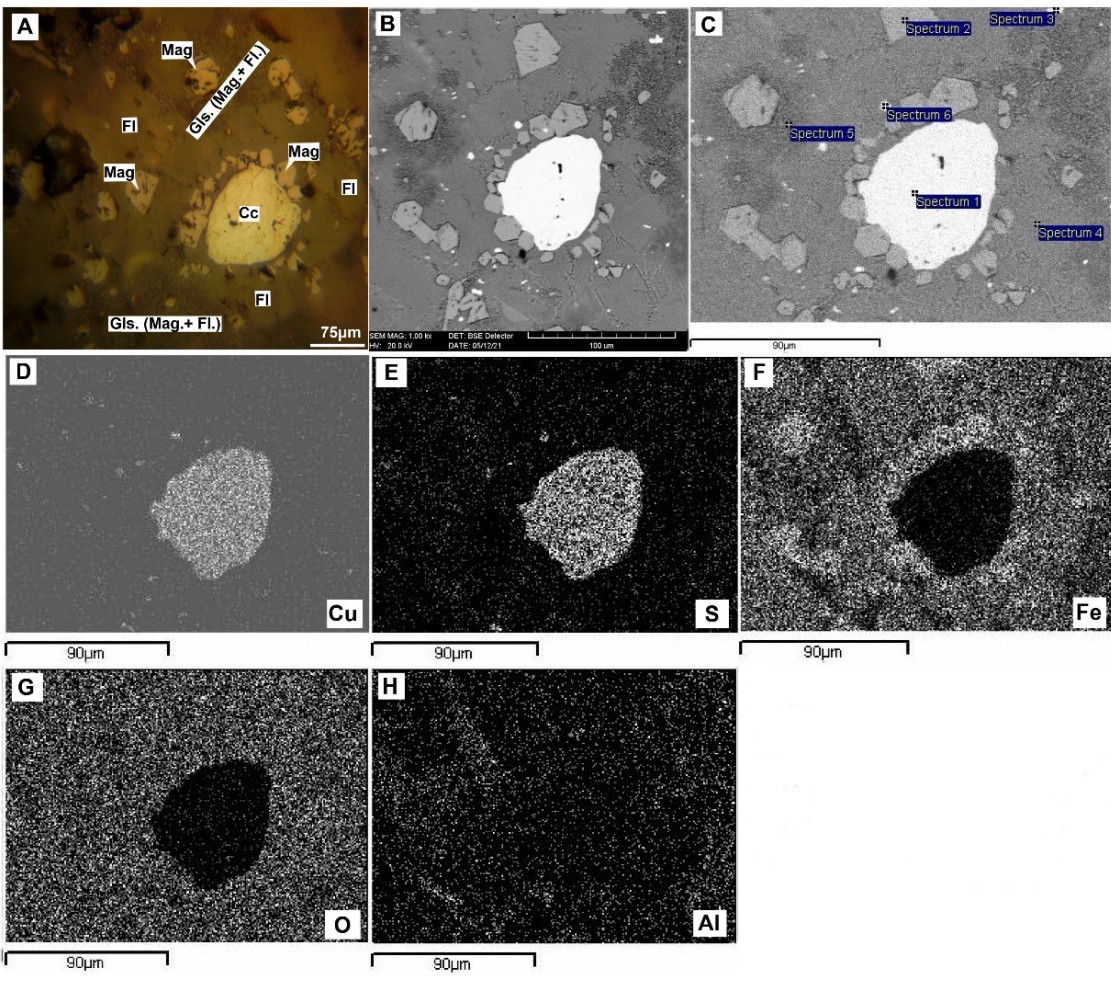

Figure 14. (A) Reflected light microphotograph, (B) back scatter electron microphotograph, (C) analytical points map, (D–H) elemental mapping images, and (I) semi-quantitative analytical data of copper-bearing phase in the converter slag sample. The composition of the main copper-bearing phase in this sample (points 1, 3, and 6) falls between chalcocite (Cc.)–yarovite (Yt.) mineral group. Other phases are magnetite (point 2) and fayalite (points 4 and 5).

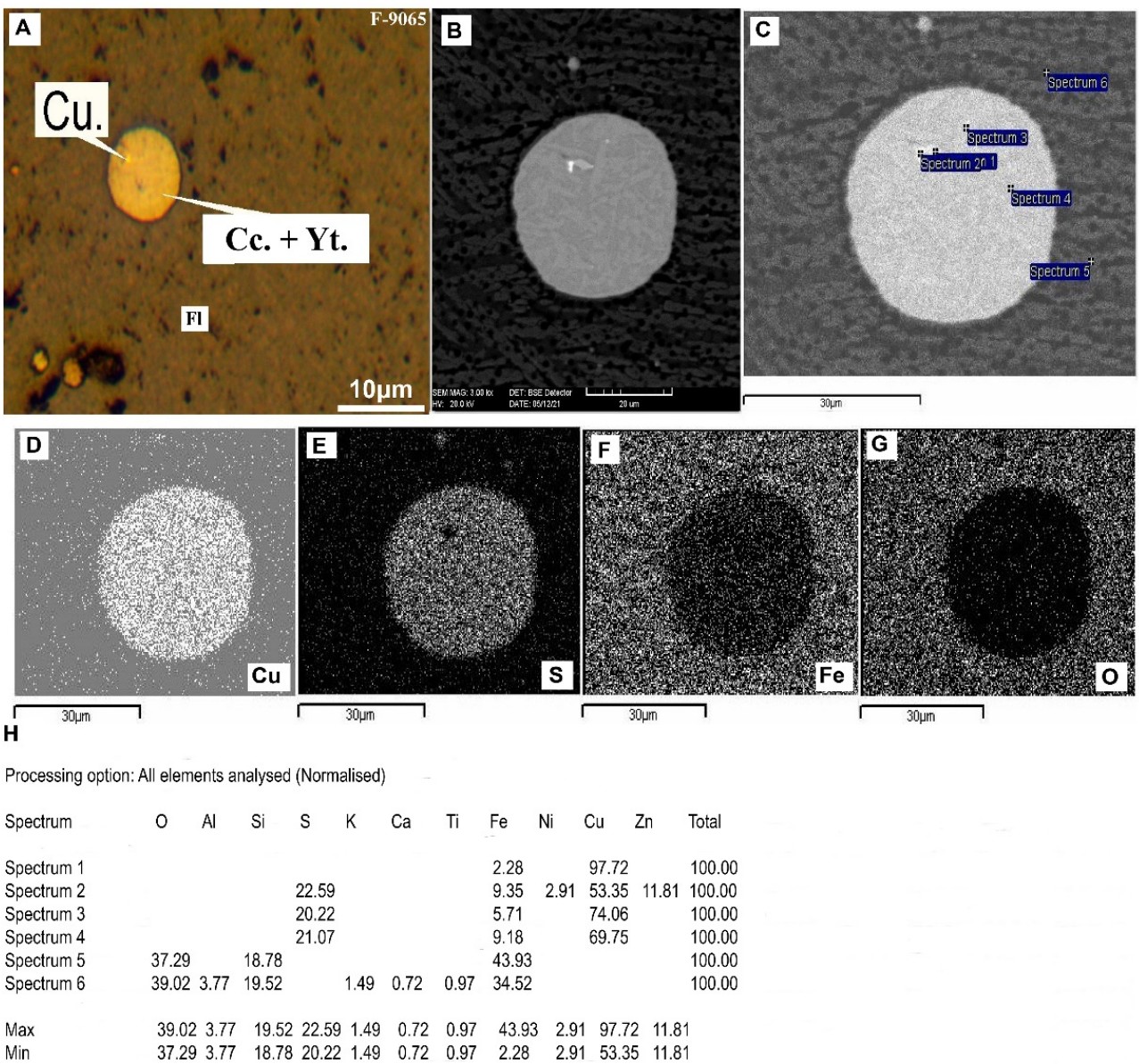

**Figure 15.** (**A**) Reflected light microphotograph, (**B**) back scatter electron microphotograph, (**C**) analytical points map, (**D–G**) elemental mapping images, and (**H**) semi-quantitative analytical data of copper-bearing phase (Cc = chalcocite and Yt = yarovite) in the flash slag sample. The composition of the main copper-bearing phase in this sample (points 1, 3, and 4) falls between chalcocite (Cc.)–yarovite (Yt.) mineral group.

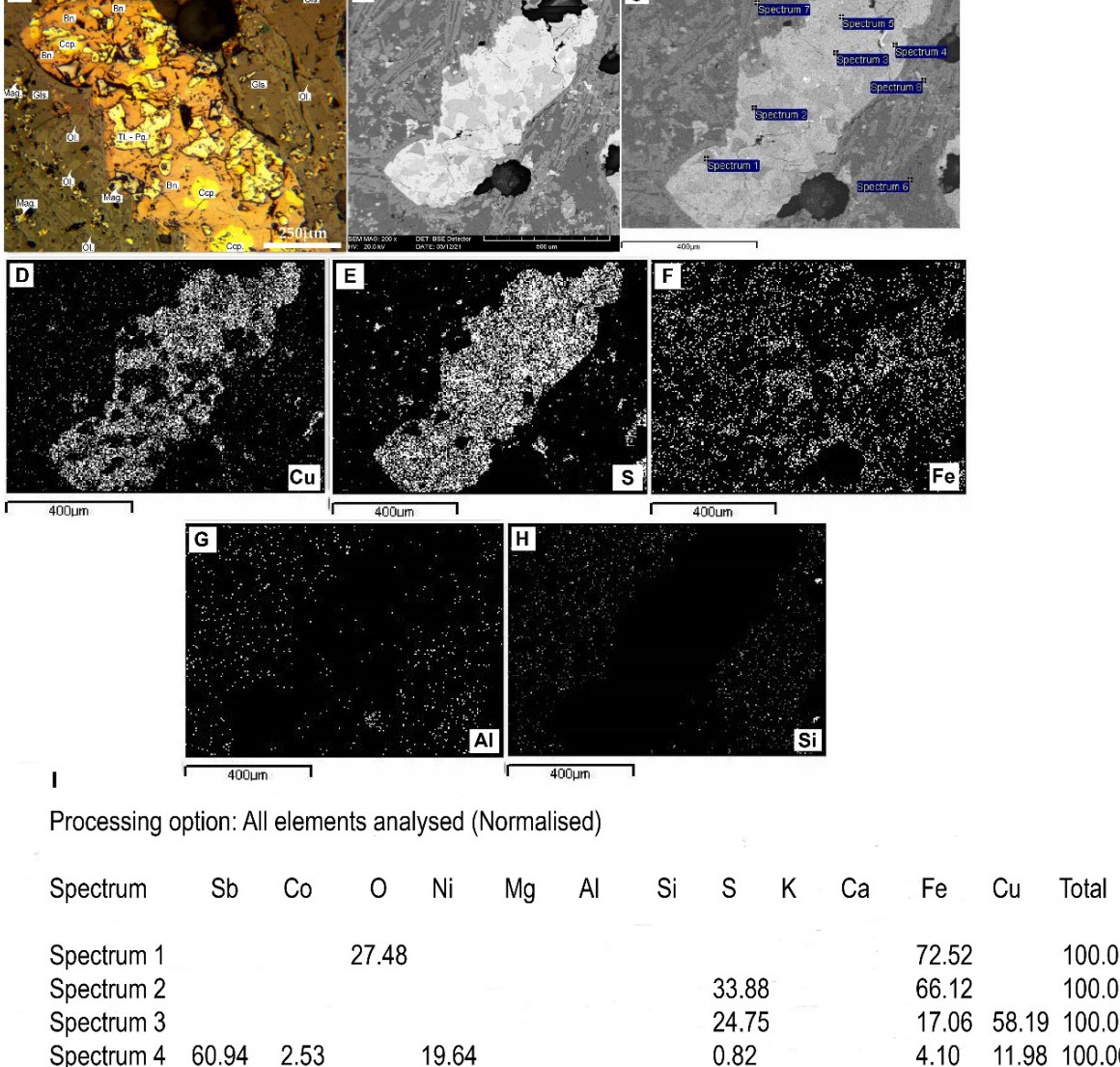

| Spectrum | Sb | Co | O | Ni | Mg | Al | Si | S | K | Ca | Fe | Cu | Total |
|---|---|---|---|---|---|---|---|---|---|---|---|---|---|
| Spectrum 1 | | | 27.48 | | | | | | | | 72.52 | | 100.00 |
| Spectrum 2 | | | | | | | | 33.88 | | | 66.12 | | 100.00 |
| Spectrum 3 | | | | | | | | 24.75 | | | 17.06 | 58.19 | 100.00 |
| Spectrum 4 | 60.94 | 2.53 | | 19.64 | | | | 0.82 | | | 4.10 | 11.98 | 100.00 |
| Spectrum 5 | | | | | | | | 30.34 | | | 39.92 | 29.74 | 100.00 |
| Spectrum 6 | | | 40.0 | | | 3.03 | 20.97 | 0.44 | 1.52 | 10.05 | 23.90 | | 100.00 |
| Spectrum 7 | | | 36.1 | | 0.70 | 0.50 | 15.21 | | 1.90 | 45.58 | | | 100.00 |
| Spectrum 8 | | | | | | | | 24.23 | | | 19.55 | 56.22 | 100.00 |

All results in Weight Percent

**Figure 16.** (**A**) Reflected light microphotograph, (**B**) back scatter electron microphotograph, (**C**) analytical points map, (**D**–**H**) elemental mapping images, and (**I**) semi-quantitative analytical data of copper-bearing phase in the reverberator slag sample. The composition of the main copper-bearing phase in this sample (points 3, 5, and 8) falls between bornite (Bn.)–idaite and chalcopyrite–cubanite mineral groups. Other phases are troilite (Tl.)–pyrrhotite (Po.) (point 2), magnetite (Mag.) (point 1), fayalite (Fl.) and pyroxene (points 6 and 7). Point 4 is probably an Sb-Ni-Cu-Fe-Co alloy phase.

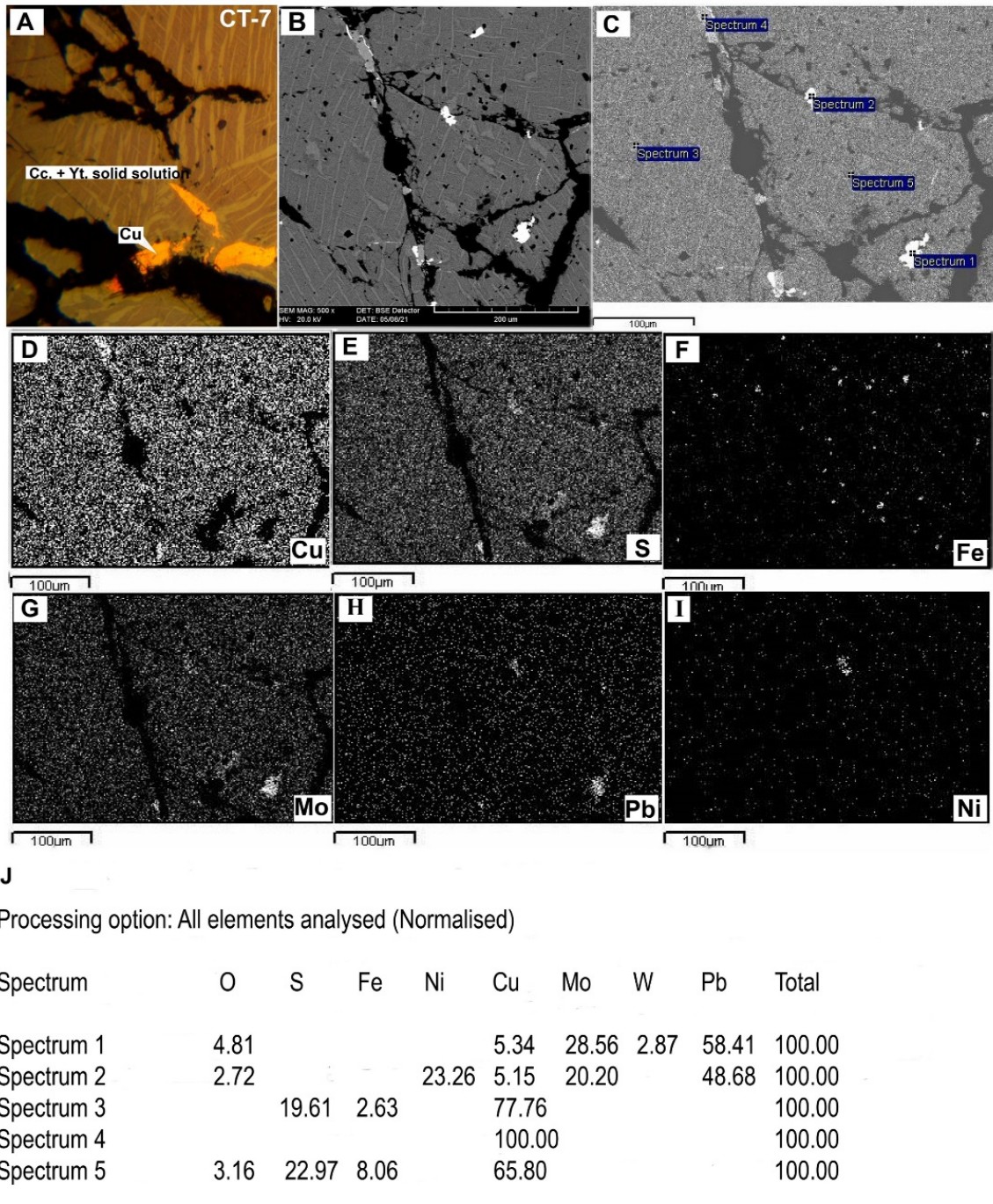

| Spectrum | O | S | Fe | Ni | Cu | Mo | W | Pb | Total |
|---|---|---|---|---|---|---|---|---|---|
| Spectrum 1 | 4.81 | | | | 5.34 | 28.56 | 2.87 | 58.41 | 100.00 |
| Spectrum 2 | 2.72 | | | 23.26 | 5.15 | 20.20 | | 48.68 | 100.00 |
| Spectrum 3 | | 19.61 | 2.63 | | 77.76 | | | | 100.00 |
| Spectrum 4 | | | | | 100.00 | | | | 100.00 |
| Spectrum 5 | 3.16 | 22.97 | 8.06 | | 65.80 | | | | 100.00 |
| | | | | | | | | | |
| Max | 4.81 | 22.97 | 8.06 | 23.26 | 100.00 | 28.56 | 28.56 | 58.41 | |
| Min | 2.72 | 19.61 | 2.63 | 23.26 | 5.15 | 20.20 | 2.87 | 2.87 | |

Processing option: All elements analysed (Normalised)

All results in Weight Percent

**Figure 17.** (**A**) Reflected light microphotograph, (**B**) back scatter electron microphotograph, (**C**) analytical points map, (**D–I**) elemental mapping images, and (**J**) semi-quantitative analytical data of copper-bearing phase in the converter slag sample. The composition of the main copper-bearing phase in this sample (points 3, 4, and 5) falls between chalcocite (Cc.)–yarovite (Yt.) and bornite–idaite mineral groups. Points 1 and 2 are probably wulfenite (PbMoO4) containing Ni-Cu-W impurities.

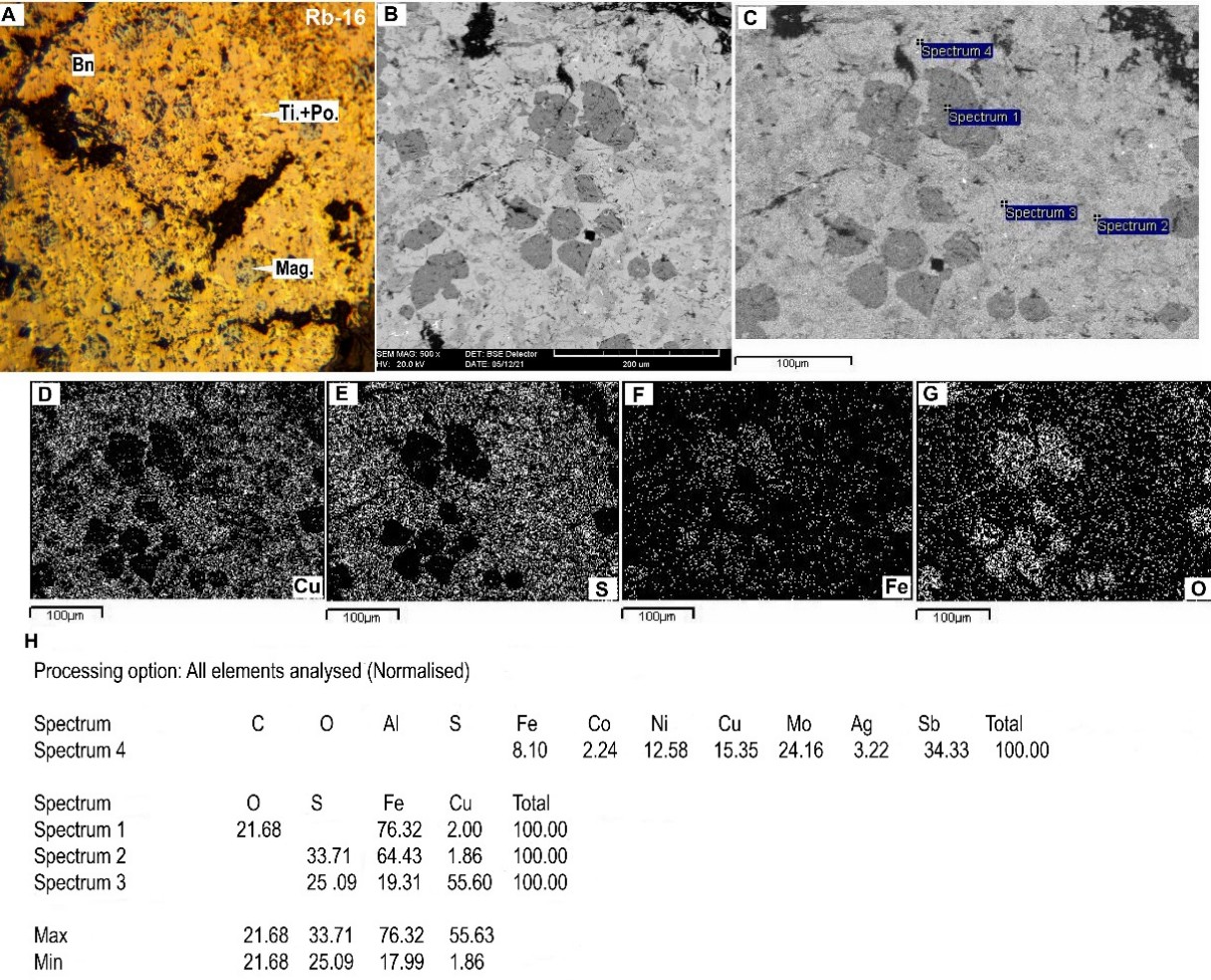

**Figure 18.** (**A**) Reflected light microphotograph, (**B**) back scatter electron microphotograph, (**C**) analytical point map, (**D–G**) elemental mapping images, and (**H**) semi-quantitative analytical data of copper-bearing phase in reverberator slag sample. The composition of the main copper-bearing phase in this sample (point 3) falls between bornite (Bn.)–idaite mineral group. Other phases within copper-bearing phase are magnetite (point 1), troilite (Tl.)–pyrrhotite (Po.) (point 2), and Sb-Mo-Cu-Ni-Fe-Ag-Co alloy (point 4).

Summing up, the compositional variation of entrained copper phases in the studied slags is by far greater in the slag samples from the reverberator and flash smelting furnace than in the converted slag samples. They can be described as $Cu\pm Fe$ sulfides with a wide range of compositions, ranging from chalcopyrite, bornite, and cubanite to Cu-Fe-S isomorphs containing solid solution lamellar intergrowths and phase change textures.

### 3.4. Slag Grinding and Liberation of the Copper-Bearing Particles

The relative hardness of the slag samples and its role in the liberation of copper-bearing particles after slag grinding was obtained through the percentage number of particles distributed in each sized fraction and variation of copper content with decreasing the particle size of the ground slag. The first mode of grinding of the studied slags showed that converter and flash slags have a relatively higher hardness than reverberator ones because the cumulative percentage of particles in the coarse-grain fractions is higher than those accumulated in the fine-grain fractions (Figure 19). Furthermore, the copper content variations along with decreasing the particle size have been different in the high- and low-Cu grade ground slag samples. In the high-Cu grade slags (i.e., <2.5 wt%), copper content

increases with decreasing the particle size, while there is no significant increase in the copper content along with particle size decreasing in the low-grade slags (i.e., <1 wt%) (Figure 20).

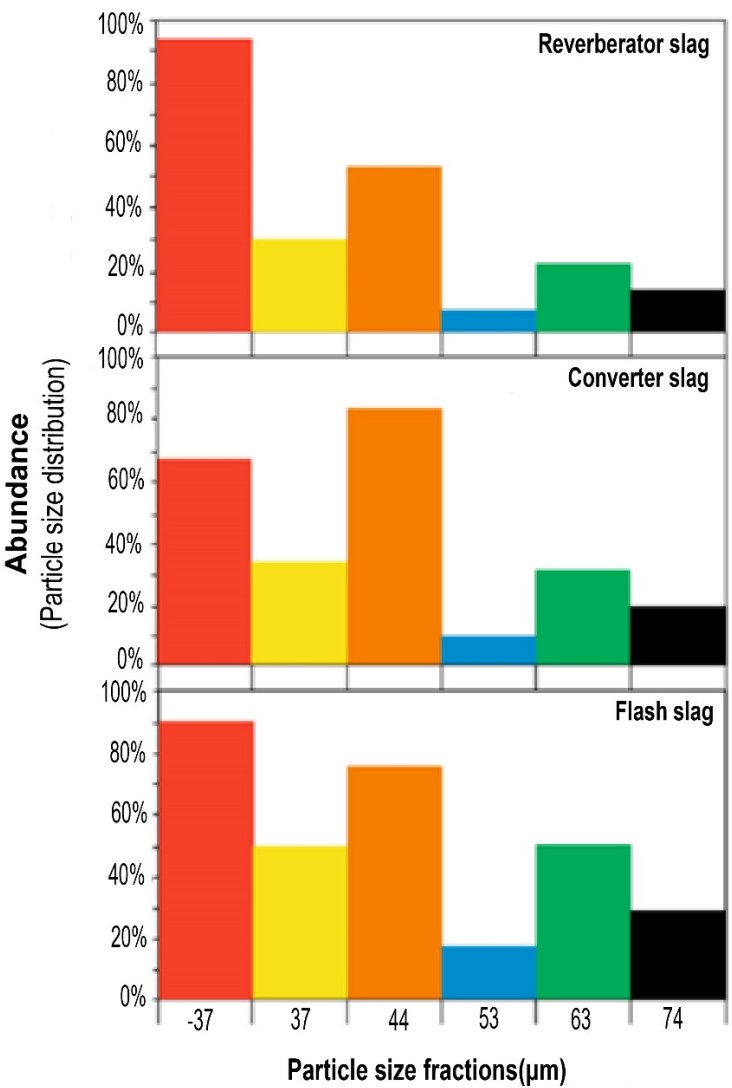

**Figure 19.** Comparison of particle size distribution in the first mode grinding of the bulk slag samples.

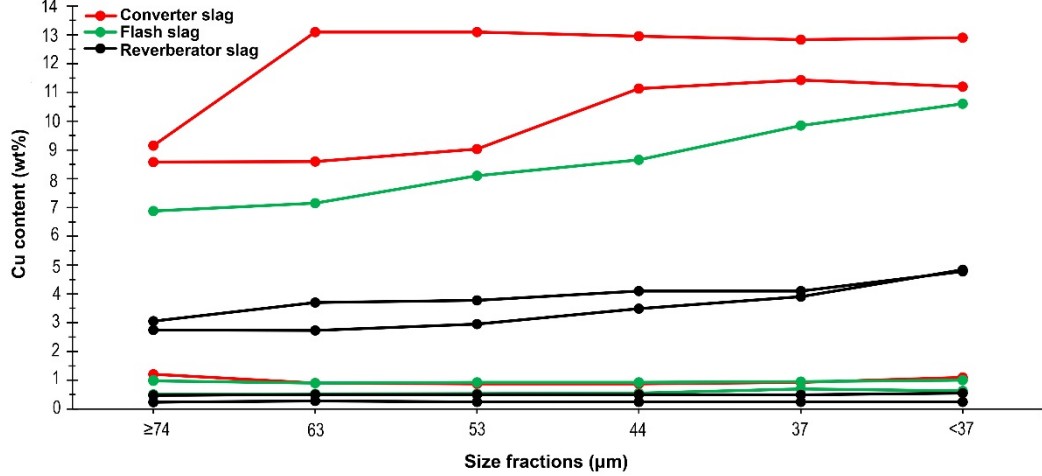

**Figure 20.** Copper content variations with decreasing the particle size in the high- and low-Cu grade ground slags.

In the second mode of grinding, in which the representative slag samples of both high- and low-Cu grades were ground over four time periods of grinding to reduce their particle size to 44 μm, size fraction analysis together with copper content assay and determination of liberation degree of copper-bearing particles in all fractions showed the better response of the reverberator and flash slags than converter ones to grinding operation (Figure 21). In the reverberator and flash slags, there is a continuous increase in the liberation degree of copper-bearing particles and copper content (wt%) toward fractions containing a higher percentage of particles <44 μm (Figure 21A–C). On the contrary, such an obvious trend is not observed in the converter slag, especially in the liberation degree of copper-bearing particles with increasing the percentage of particles <44 μm despite the increase in the copper content (Figure 21D). Microscopic observations confirmed the better response of reverb and flash slags to grinding operations to obtain higher degrees of liberation of copper-bearing particles (Figures 22–24) compared to converter slags (Figure 25).

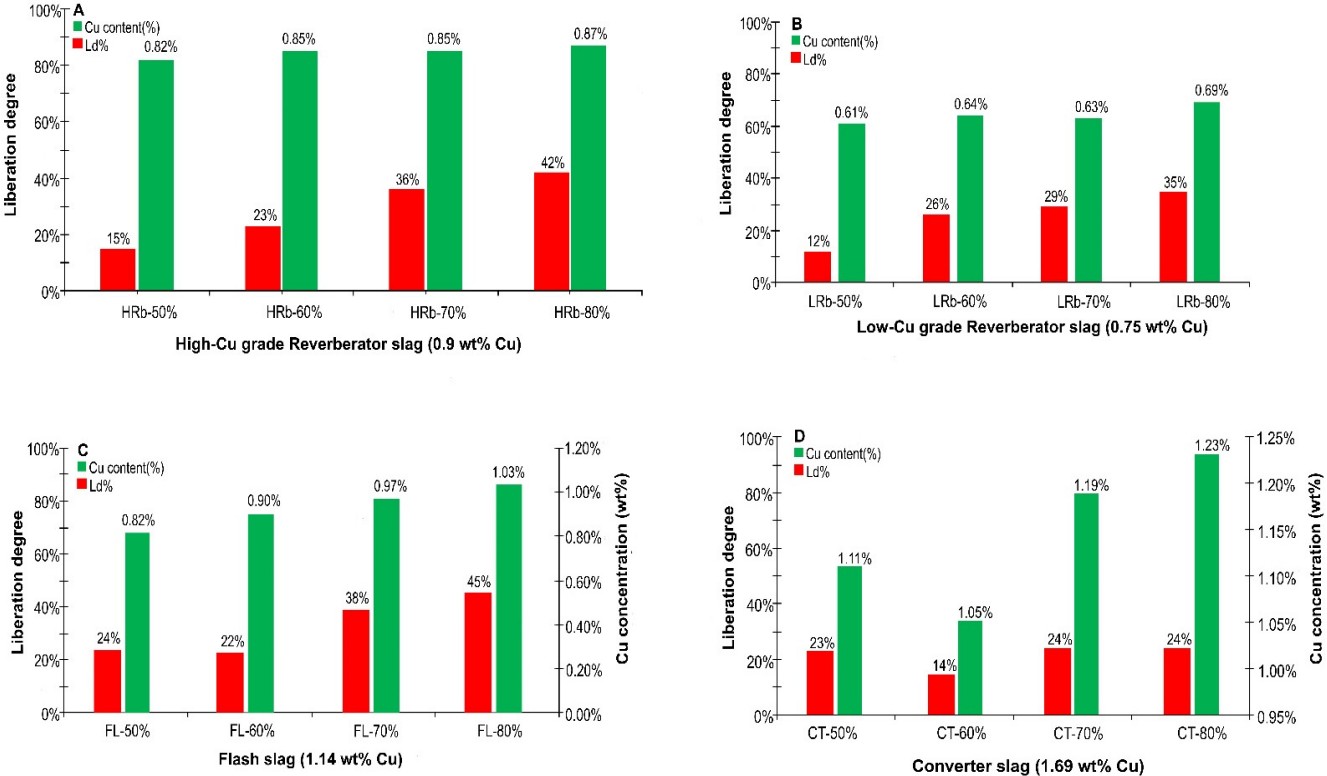

**Figure 21.** Comparative plots showing variations of copper content and liberation degree of the copper-bearing particles in the ground slags containing different percentages of particles < 44 μm.

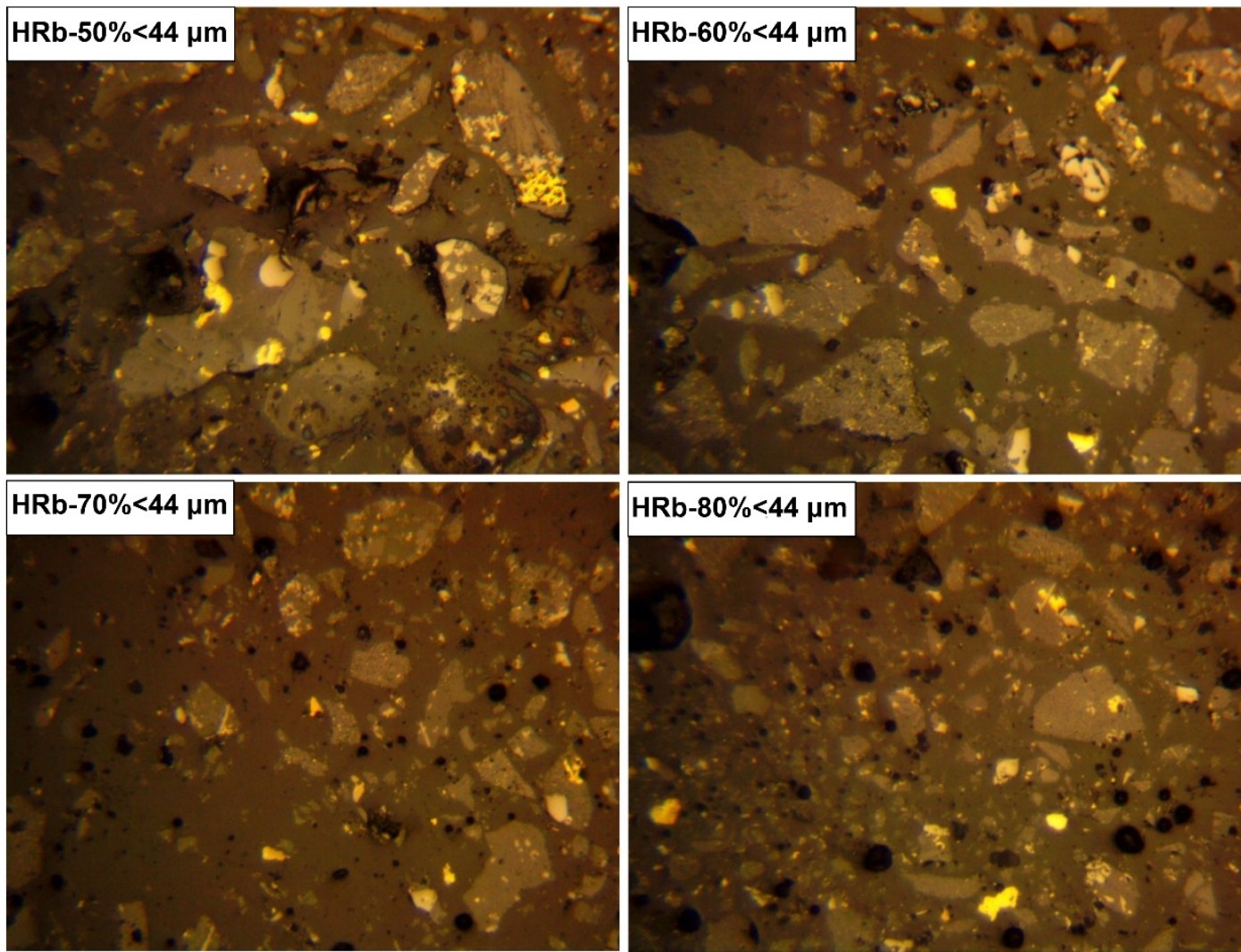

**Figure 22.** Reflected light microphotographs (20× magnification) comparing the particle size and liberation rate of copper-bearing particles in four fractions with different percentage of particles < 44 μm in the high-Cu grade ground samples of the reverberator slag (i.e., HRb = 0.9 wt% Cu). Noted that the copper particles (characterized by yellow in color and a higher reflectance than non-valued phases) are locked in the ground sample containing 50% of particles passing 44 μm (i.e., HRb-50% < 44 μm microphotograph) and their liberation as fine particles in the slag with a higher percentage of particles passing 44 μm (i.e., HRb-60%, HRb-70%, and HRb-80% < 44 μm microphotographs).

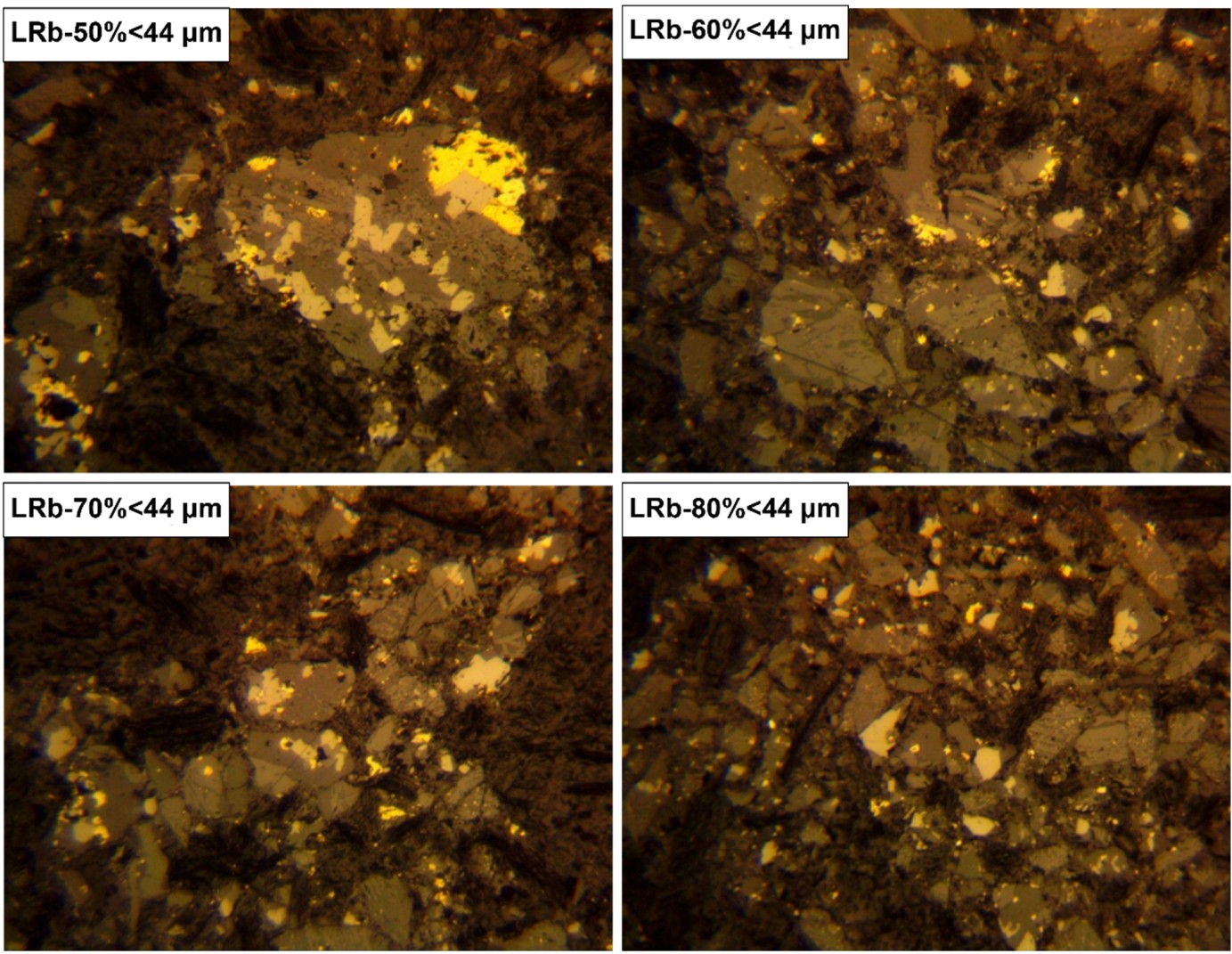

**Figure 23.** Reflected light microphotographs (20× magnification) comparing the particle size and liberation rate of copper-bearing particles in four fractions with different percentage of particles < 44 μm in the low-Cu grade ground samples of the reverberator slag (i.e., LRb = 0.75 wt% Cu). Noted that the copper particles (characterized by yellow in color and a higher reflectance than non-valued phases) are locked in the ground sample containing 50% and 60% of particles passing 44 μm (i.e., LRb-50% and LRb-60% < 44 μm microphotographs) and their liberation as fine particles in the slag with a higher percentage of particles passing 44 μm (LRb-70% and LRb-80% < 44 μm microphotographs).

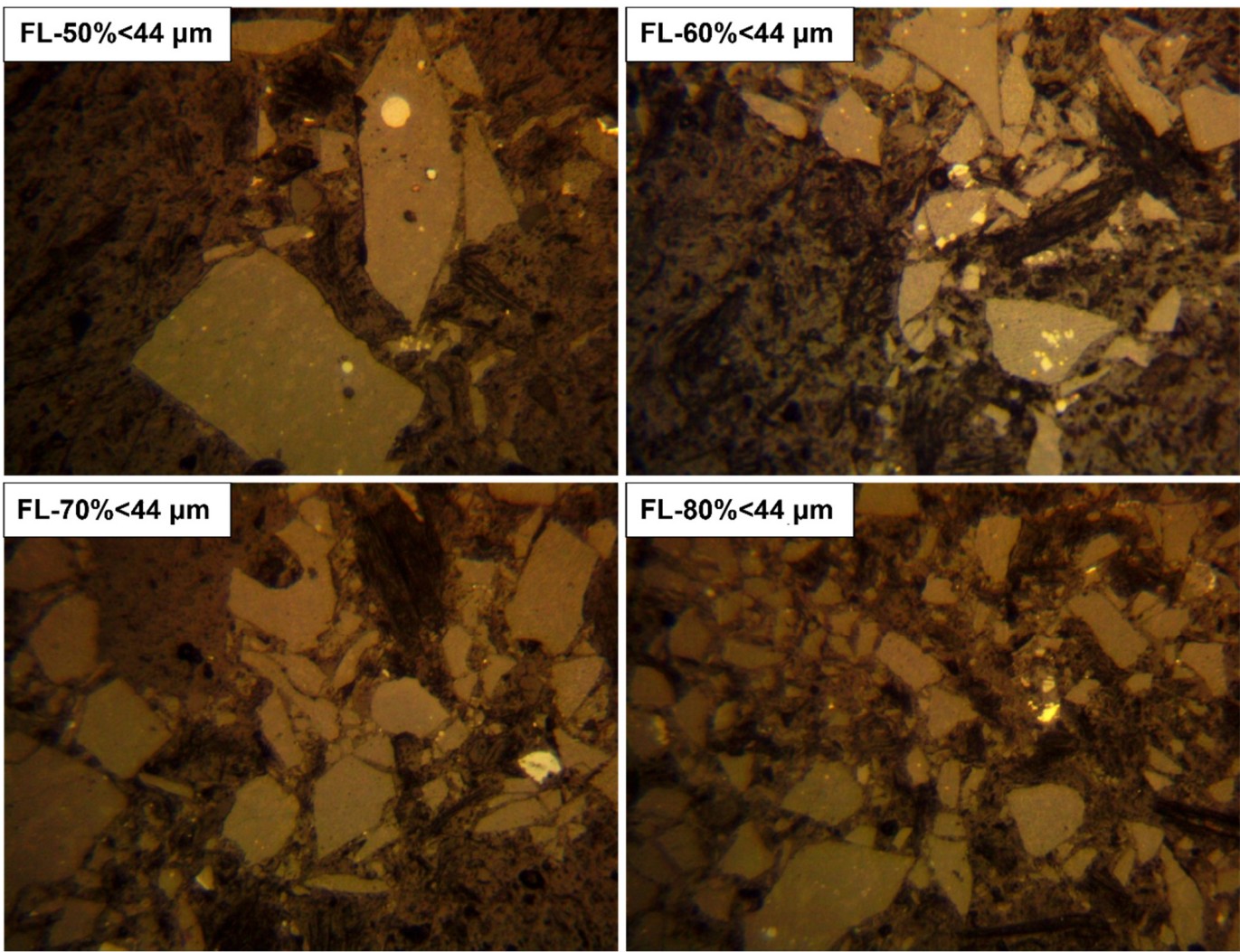

**Figure 24.** Reflected light microphotographs (20× magnification) comparing the particle size and liberation rate of copper-bearing particles in four fractions with different percentage of particles < 44 μm in the ground samples of the flash slag (i.e., FL = 1.14 wt% Cu). Noted that the copper particles (characterized by yellow in color and a higher reflectance than non-valued phases) are locked in the ground sample containing 50% and 60% of particles passing 44 μm (FL-50% and FL-60% < 44 μm microphotographs) and their liberation as fine particles in the slag with a higher percentage of particles passing 44 μm (FL-70% and FL-80% < 44 μm microphotographs).

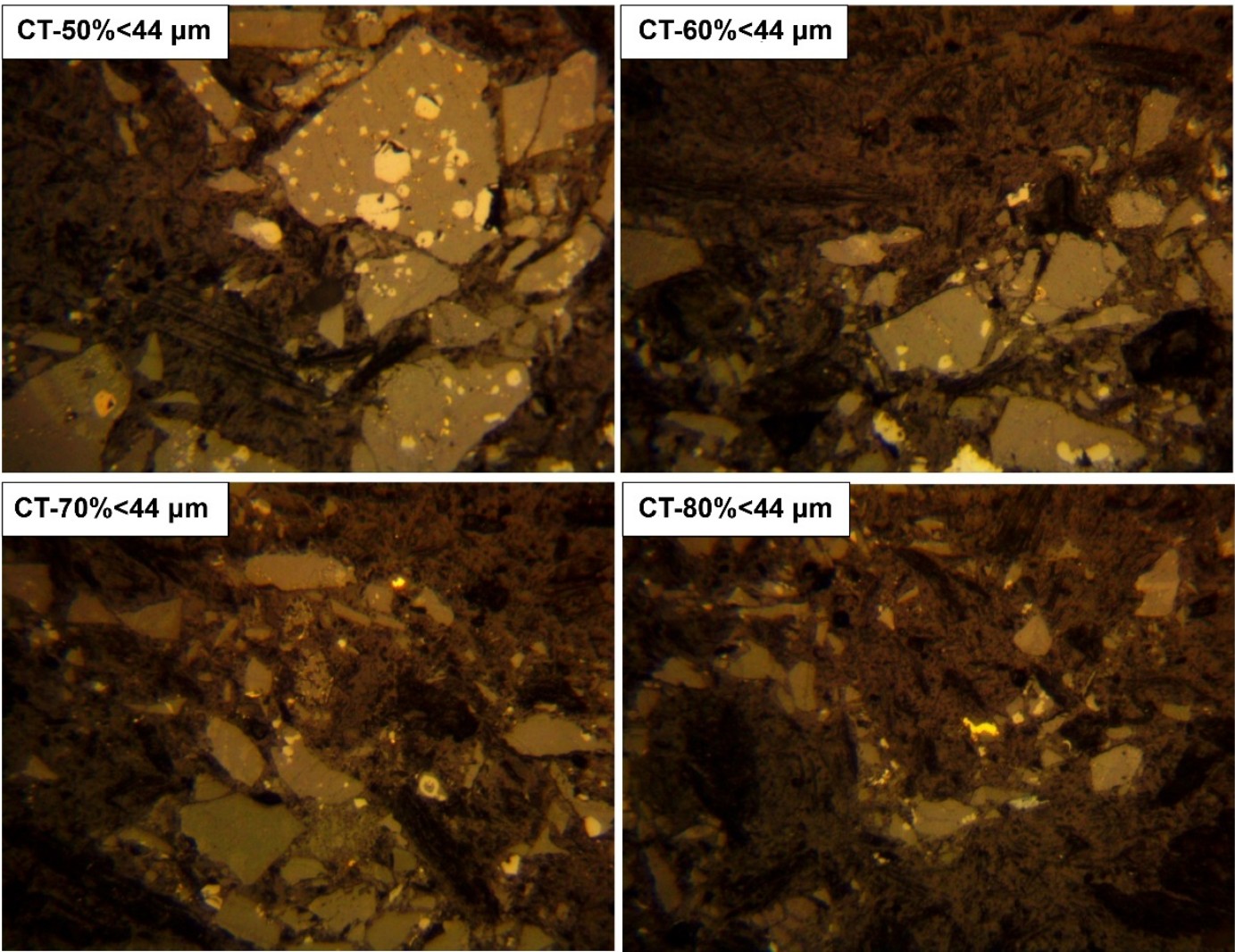

**Figure 25.** Reflected light microphotographs (20× magnification) comparing the particle size and liberation rate of copper-bearing particles in four fractions with different percentage of particles < 44 μm in the ground samples of the converter slag (i.e., CT = 1.69 wt% Cu). Noted that the copper particles (characterized by yellow in color and a higher reflectance than non-valued phases) are locked in the ground sample containing 50% of particles passing 44 μm (i.e., CT-50% < 44 μm microphotograph) and their liberation as fine particles in the slag with a higher percentage of particles (80%) passing 44 μm (i.e., CT-80% < 44 μm microphotograph).

## 4. Discussion

### 4.1. Copper Losses into the Slag

The copper-bearing phases in the studied slag samples have shown different composition, shape, size, and various textures, which allows us to identify the type of copper losses into the slag and discuss how the different textures are formed. The disseminated grain texture was one of the most important textures observed for copper-bearing phases in the studied slags. Copper sulfide and metallic copper phases in the form of rounded grains with dimensions mostly less than 50 μm disseminated in the silicate glassy matrix, mostly occupying the gaps between the large elongated skeletal crystals of fayalite although, can be attributed to the mechanical copper losses into the slag. Furthermore, the spherical, ovoid, and teardrop shapes of fine to coarse grains (from 20 μm up to 200 μm) of copper matte which was scattered in the hypocrystalline texture of flash and converter slags can be attributed to the mechanical losses of copper into the slags, as previously reported in the literature. Mechanical entrapment of matte in the slag samples can be due to the effect

of slag viscosity related to the excessive formation of magnetite in slag [7,8,19–26]. An increase in the Fe concentration in the slag leads to the separation of magnetite crystals, which causes heterogeneity of silicate melt and sudden increase in its viscosity [27] inhibiting the slag–matte separation and so increasing the mechanical losses of copper into the slag [8,23,28]. This fact could explain the higher abundance of copper-bearing particles found in the magnetite-rich slag samples from the converter compared to flash slags. Although due to the lack of quantitative mineralogical data, it is not possible to determine the relationship between the amount and type of copper losses and the magnetite content of the slag samples, according to data obtained from the optical microscopy the converter slags that had higher proportions of coarse and euhedral to subhedral grains of magnetite compared to flash ones showed more mechanical losses of copper.

In contrast to the mechanical entrapment of matte in the slag, the chemical losses of copper into the slag can be introduced by the presence of chalcopyrite–bornite mineral pair particles and also chalcopyrite–bornite veinlets both with transformation or phase change texture. In these cases, coarse grain chalcopyrite (between 100 and 150 µm) from the rim or along the fracture and cleavage has been transformed to bornite, or in some cases, transformation of bornite to chalcocite is observed. The transformation or phase change texture was mostly observed in reverberator slag and to some extent in flash slag samples. The origin of the chemical losses of copper into the slag and the formation of this texture actually begins with the mechanical losses of the molten copper into the slag and its nucleation. Due to the changes in the physicochemical conditions of the furnace environment, the initial nucleus grows, and changes phase occurs [7,8,29,30]. Blowing more air into the furnace and increasing oxygen fugacity at high temperatures likely prompted the transformation of chalcopyrite to bornite, which was subsequently bornite transformed to chalcocite by increasing oxygen fugacity as a result of blowing more air into the flash and converter furnaces. Theoretical and experimental works showed that crystallization of copper sulfides (i.e., chalcopyrite, bornite and chalcocite) from molten slag begins with a progressive decrease in temperature until 435 °C and increasing oxygen fugacity in furnace environment [31–34]. In the present study, chalcopyrite was found in reverberator furnace slags in two forms: single-phase particles and, in many cases, chalcopyrite–bornite mineral pair particles with the transformation or phase change texture to bornite. The transformation texture and phase change that has occurred along the cleavages and as well as at the rim of the chalcopyrite can be related to increasing temperature and/or increasing oxygen fugacity during slag production, causing the transformation of the tetragonal crystalline structure (chalcopyrite) to the cubic (bornite) at a higher temperature (557 °C) [31–35]. The transformation of bornite from the rims and cleavages to chalcocite ($Cu_2S$) is also due to the further increase in the oxygen fugacity of the furnace environment, which transformation of the cubic crystalline structure (bornite) to cubic (chalcocite) may have happened at a temperature exceeding 435 °C [31–34]. Moreover, the presence of chalcopyrite–bornite veinlets in the reverberator slag samples confirm the entry of the molten copper into the slag, its flow, and then crystallization of copper sulfides in the form of veinlets from the molten copper whose temperature is decreasing to 547 °C.

Solid solution or lamellar intergrowths are the other main textures observed in spherical matte particles in the studied flash and converter slag samples. Based on SEM analytical data, this texture was observed between minerals of the yarrovite ($Cu_9S_8$)–chalcocite ($Cu_2S$) mineral group that contains droplets or veinlets of metallic copper ($Cu^0$). Formation of this texture has been attributed to exsolution and crystallization upon cooling and probably increasing the oxidation time during slag–matte separation process [31,36,37].

Formation of pyrrhotite–troilite mineral pair coexist with chalcopyrite–bornite mineral pair in the studied reverberator slag samples indicates the low fugacity of oxygen in the furnace environment at the time of matte-slag separation. Theoretical and experimental works have shown the crystallization of pyrrhotite–troilite mineral pair from the cooling of molten slag from 1000 °C to 742 °C [31,38,39]. The coexistence of hematite ($Fe_2O_3$) with magnetite ($Fe_3O_4$) in the converter slags can be interpreted in three ways: (a) changing

the oxidation state of the molten slag, (b) its oxygen fugacity heterogeneity related to the addition of oxygen, and/or (c) equilibrium redox conditions compatible with the formation of both Fe-oxide phases [19,40,41].

### 4.2. Applications to Copper Recovery

Flotation is one of the processes used to recover copper phases from the copper slag in the world, which is also selected as a process for slag processing in the SarCheshmeh Copper Complex. Considering that the copper slags are hard materials, to recover their copper by flotation process, grinding should be carried out to obtain optimal particle size. The efficiency of slag grinding operation to achieve the maximum liberation degree of copper-bearing particles could be controlled by the texture and mineralogy of the slag, which is related to the rate of slag cooling and its solidification. The cooling rate changes the abundance and type of copper-bearing phases and affects the liberation of copper-bearing particles from the slag [15–17,42–46]. When the molten slag is cooled slowly, a dense, hard, and crystalline slag forms. Under such conditions, the dimensions of the copper-bearing particles increase, which leads to an increase in the total copper content of the slag which will cause the liberation of more particles of copper-bearing phases during grinding. As a result, the liberation degree of copper-bearing particles is obtained in larger quantities. In contrast, under rapid cooling rate, an amorphous and glassy slag forms. Because copper-bearing phases have not had time to grow enough are primarily encapsulated in the form of fine particles in the rapidly crystallizing glass matrix which are not easily liberated by grinding [15,40,47–49].

Due to leaving the molten slag in landfills and relatively slow cooling in the air (air-cooled mode), the studied slags have mostly a fully to semi-crystalline texture (containing 60–95 vol% crystals) with an average of 5–40% amorphous glassy matrix. From the value recovery point of view, reverberator slag samples have shown differences in crystal–glass matrix ratio, and also in composition, abundance, and mode occurrence of copper-bearing phases when compared to the converter and flash slag samples. The reverberator slag samples have shown semi- to fully crystalline texture with crystal–glass matrix ratio from ≤1 to >1, respectively. In semi-crystallized reverberator slags, copper-bearing phases as fine- to medium-grain particles are distributed within the silicate glassy matrix, while fully crystallized reverberator slags with crystal–glass matrix ratio > 1 have shown medium- to coarse-grain copper-bearing particles that fill the hollows and gaps of the skeletal crystals of fayalite and euhedral to subhedral crystals of magnetite, indicating their crystallization from a slowly cooled Cu-saturated molten slag. This textural pattern suggests that a considerable amount of entrained copper-bearing phase particles could be efficiently recovered from such solidified slag by conventional or fine grinding (particle size between 74 and 44 μm) and froth flotation. The flash slags have shown both semi- to fully crystalline textures similar to the reverberator slags. In semi-crystallized slags, copper-bearing phases including chalcopyrite and bornite with medium- to coarse-grain particles are scattered both within the silicate glassy matrix and fill spaces between large crystals of fayalite and magnetite. In contrast, copper-bearing phases in the hypocrystalline flash and converter slags are present in the form of irregular- to spherical-shape of fine- to coarse-grain high-grade matte particles are closely embedded in the fully crystallized fayalitic matrix, revealing that even ultrafine grinding cannot result in efficient recovery of copper. The liberation degree of the copper-bearing particles is one of the most important factors controlling the efficiency of the value recovery process of the slags. Based on the literature review, experimental works show that copper-bearing particles in slag reach the highest degree of liberation only when the slag is ground to reduce the particle size to 44 μm, which will be resulted in efficient recovery of copper [17,18,50–52]. The first mode of grinding tests under the same time of grinding (20 min.) in our study showed that increasing portions of fine (≤44 μm) to ultrafine fractions (<37 μm) in the initially high-Cu grade slags will lead to an increase in the Cu content of the ground slags but such a relationship was not observed in the originally low-grade ones. This could be related to the

abundance of fine copper-bearing particles trapped in the silicate glassy matrix or inside the main crystalline phases in the originally low-grade copper slags which even ultrafine grinding could not liberate copper particles. In contrast, our result showed that the initially high-grade copper slags containing fine- to coarse-sized particles of copper-bearing phases even at relatively moderate grinding rates (mostly particle size $\leq$ 44 microns) reach the optimal liberation degree of copper-bearing phases, leading to their effective and efficient recovery. The second mode of grinding tests under different grinding times with the aim of achieving the optimal particle size and liberation degree of the copper-bearing particles showed a better response of reverberator and flash to the grinding compared to the converter ones. Such milling response led to increasing the liberation degree of copper-bearing particles and then the increase in copper content due to increasing portions of fine grinding (80% particle size < 44 $\mu$m) during 70 min of grinding operation. In contrast, although the increase in fine fractions has been associated with the increase in copper content, it has not increased the liberation degree of copper-bearing particles in the converter slag. This is the reason for the presence or remaining of ultra-fine particles in the fully crystalline matrix of the converter slags probably related to the greater resistance of converter slags to grinding due to their higher hardness compared to flash and reverberator slags. Multi-stage or regrinding to reduce the particle size to <37 $\mu$m will result in efficient recovery of copper from the converter slags.

## 5. Conclusions

The mineralogical studies showed that the residual copper in the slags produced by reverberator-, flash-, and converter-smelting furnaces from the SarCheshmeh smelter plant in Iran, is largely in the form of sulfide minerals i.e., close to the theoretical composition of common copper sulfide minerals such as chalcocite ($Cu_2S$), chalcopyrite ($CuFeS_2$), bornite ($Cu_5FeS_4$), and metallic copper, which are scattered in the semi- to a fully crystalline matrix consisting fayalite and magnetite with/without an amorphous silicate glass phase. The slags contain copper content ranging from 0.78 to 18.75 wt%. The reverberator slags show lower Cu content (0.34–5.75 wt% with an average of 1.77 wt%) compared to the flash (up to 18.75 wt% with an average of 2.44%) and the converter slags (1.21–18.65 wt% with an average of 4.65 wt%). Due to this high copper content, the production of copper sulfide concentrates from these slags by using the flotation process will soon begin in the SarCheshmeh copper complex.

The slag samples from the flash and reverberator furnaces showed both copper sulfide phases (chalcopyrite and bornite) and high-grade matte particles, revealing copper losses in both chemical and mechanical modes. In the reverberator slags, Cu-Fe-S phases are mainly in the form of medium- to coarse-grain particles of chalcopyrite—bornite mineral pair (100 up to 200 $\mu$m) which are easily recovered by fine grinding and froth flotation. In contrast, fine-grain particles of the metallic copper phase along with Cu-Fe-S phases scattered in the silicate glassy matrix are unlikely to recover even in ultrafine grinding operation. Therefore, they will enter the tailings during flotation. The dominant copper-bearing phases in the converter slags were chalcocite and metallic copper matte, which were mainly scattered in the form of fine- to coarse-grain spherical particles (less than 10 up to 200 $\mu$m) within the fully crystallized matrix of fayalite and magnetite. This study showed that copper losses into the converter slags are mainly as mechanically entrapped high-grade matte particles, suggesting that the copper grade of the matte plays an important role in the copper losses.

The key mineralogical properties affecting copper recovery of the slag by flotation operation are texture, size, composition, and liberation degree of copper-bearing phases. Texture includes the crystal–glass matrix ratio, mode occurrence, and distribution of copper-bearing phases. The fully crystalline slags characterized by crystal–glass matrix ratio > 1 containing moderate- to coarse-grain copper-bearing particles coexisting with main crystal phases (i.e., fayalite and magnetite) and much better milling response, will result in efficient recovery of a significant amount of copper. In contrast, the semi-crystalline slags with

crystal–glass matrix ratio $\leq 1$ respond very poorly to metal recovery due to locking the fine-to moderate-grain copper particles in the silicate glassy matrix. Grinding tests with the aim of achieving the optimal particle size and the desired liberation degree of copper-bearing particles showed that conventional ($\leq 74$ μm) to fine grinding ($\leq 44$ μm) of the originally high-Cu grade slags leads to a significant increase in the liberation degree of copper-bearing particles, leading to their easier recovery. Increasing portions of fine particles ($<37$ μm) due to re-grinding or ultrafine grinding are unlikely to increase the liberation of copper-bearing particles in the originally low-Cu grade slags but it will also cause the production of ultra-fine particles in excessive amounts which reduce the efficiency of the flotation process. Our results suggest that the highest rate of copper recovery of the slag by the flotation process will be obtained at particle size 80% passing 44 μm which has also reached the optimal liberation degree of copper-bearing particles. Apart from the slag texture and liberation degree of copper-bearing particles, the presence of copper phases with unknown behavior during the flotation process (i.e., Cu-Fe-S isomorph compounds, Fe-S isomorphs containing Cu impurity, and Cu-Fe alloy) may control the efficiency of copper recovery. The role of non-conventional copper-bearing phases in the poor recovery of the slag and also copper losses to flotation tailings is not well understood, so it is necessary to know more in this area to improve the slag flotation process worldwide.

**Author Contributions:** Conceptualization, B.S.B. and M.R.Y.; Methodology, B.S.B., S.M.N., M.M.M. and J.K.K.; Investigation, B.S.B. and S.M.N.; Data curation, B.S.B.; Writing—Original draft preparation, S.M.N.; Writing—review and editing, B.S.B.; Supervision, B.S.B.; Project administration, M.R.Y.; Funding acquisition, B.S.B. and M.R.Y. All authors have read and agreed to the published version of the manuscript.

**Funding:** This research was funded by the R&D of SarCheshmeh Copper Complex, National Iranian Copper Industries Company, Rafsanjan, Iran.

**Data Availability Statement:** All data used in the current research are presented in the main text of the manuscript, and there is no supplementary data.

**Acknowledgments:** We appreciate the support of the National Iranian Copper Industries Company. We would like to thank the Central Laboratory of SarCheshmeh Copper Complex, Iran, Iran Mineral Processing Research Center (IMPRC), ZarAzma Mahan Company, and Metallurgical Research Center of the Shahid Bahonar University of Kerman, Kerman, Iran for performing the required analyses in this study. We are thankful to the editor and two reviewers whose comments helped us improve the manuscript.

**Conflicts of Interest:** The authors declare no conflict of interest.

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
