# Peer review of "Mineralogical Properties of the Copper Slags from the SarCheshmeh Smelter Plant, Iran, in View of Value Recovery"

_minerals, doi:10.3390/min12091153_

Round 1

Reviewer 1 Report

Dear Authors,

please find attached pdf file of peer review.

Sincerely,

Author Response

Due to the relatively complete rewriting of all parts of the article, almost all of the reviewer's comments have been addressed in the revised version of the article.

Reviewer 2 Report

The article “Mineralogical properties of the copper slags from the Sar-2 Cheshmeh Smelter Plant, Iran, and its applications to value recovery by flotation operation” is based on the characterization of three slags which contain copper. The work looks at different characterization methods used to identify the copper phases and how the copper can be recovered. The methodology of the paper is very poor as it does not address a lot of the results which you report. There are a lot of corrections and comments on this in the sections.

The title talks of characterization of copper slags which you did well. It also talks of flotation as a recovery method of this copper which you did not do any experiments on. So the title is misleading in saying flotation is the recovery method for copper and also by its nature, flotation is not a recovery technique. Instead, flotation is only used to preconcentrate the copper and you need another method to recover. In addition, flotations has many aspects to consider before it can be said it is the best method to use which you did without due consideration of such parameters. Flotation also is reliant on comminution which you did not talk about in your study as well or you did not do any experiments on.

You need to mention other methods available besides flotation and why flotation is the best method to do this and actually do the experiments on it.

Author Response

Due to relatively complete rewriting of all parts of the article, almost all of the referee's comments have been answered in the edited version of the article.

Reviewer 3 Report

The paper mainly investigated the mineralogical Properties of three types of Copper Slags and on this basis talked about its applications to value recovery by flotation operation. Overall, the paper is well written and discussed although relevant topics regarding the mineralogical properties of copper slags have been studied by many researchers previously. The paper can be considered for publication if following comments can be well addressed.

(1)    The authors need to check the spelling and some other linguistic errors in the article. Such as Page 6, Line 208-“the flash…” should be “the flash slags”; Page 22, Line 713 “the studied slags”; Page 22, Line 723 “the falsh” should be “the flash”, etc.

(2)    Section 3.1, Page 3, Lines 119-121: It's hard to say from the Fig. 2 that the flash slag D and E are less porous than B and C. From my side, I draw the opposite conclusion. Therefore, if possible, the authors can conduct the porosity measurement for the slags.

(3)    Page 4, Lines 128-129: Why is the Bond Work Index (BWI) of copper slags much higher than that of copper ores? The BWI of which type of copper slag is reported 26.8 kWh/t?

(4)    In terms of Fig. 3, I would like suggest authors calculating and providing the mean particle size of ground slags. In this case, it’s easier to compare.

(5)    What's the analytical method for measurement of slag density in Fig. 4? Why does each type of copper slag possess a wide range of density?

(6)    The quality of the figures 7, 14 and 16 are too poor to read the information.

(7)    Copper slags are still one of world-known difficult ores. Based on the mineralogical characteristics of three types of copper slags, what are the key obstacles affecting the efficient recovery of metal values?

(8)    Besides the flotation, any suggestions for pyrometallurgical methods of metals recovery from copper slags?

Author Response

Due to the relatively complete rewriting of all parts of the article, almost all of the referee's comments have been answered in the edited version of the article.

Round 2

Reviewer 1 Report

The manuscript with edited title «Mineralogical properties of the copper slags from the SarCheshmeh smelter plant, Iran, - in view of value recovery» by Behnam Shafiei Bafti, Saeed Mohamadi Nasab, Mohamad Reza Yarahmadi, Mohammad Mahmoudi Maymand, Javad Kamalabadi Khorasani was submitted was second review.

The presented version of the corrected article is a complete scientific study and from my point of view will be very interesting to the reader. All remarks are taken into account, which makes the manuscript easier for readers to perceive. The manuscript can be published in the open press in present form.

Reviewer 2 Report

My comments have been addressed to the best of my knowledge.

Reviewer 3 Report

Most comments are well addressed. The paper can be considered for publication.